# Eco-evolutionary strategies for relieving carbon limitation under salt stress differ across microbial clades

Yang Dong[1,2,3], Ruirui Chen [1,3] ✉, Emily B. Graham [4,5] ✉, Bingqian Yu[1,3], Yuanyuan Bao [1,3], Xin Li[1], Xiangwei You[2] & Youzhi Feng [1,3,6]

With the continuous expansion of saline soils under climate change, understanding the eco-evolutionary tradeoff between the microbial mitigation of carbon limitation and the maintenance of functional traits in saline soils represents a significant knowledge gap in predicting future soil health and ecological function. Through shotgun metagenomic sequencing of coastal soils along a salinity gradient, we show contrasting eco-evolutionary directions of soil bacteria and archaea that manifest in changes to genome size and the functional potential of the soil microbiome. In salt environments with high carbon requirements, bacteria exhibit reduced genome sizes associated with a depletion of metabolic genes, while archaea display larger genomes and enrichment of salt-resistance, metabolic, and carbon-acquisition genes. This suggests that bacteria conserve energy through genome streamlining when facing salt stress, while archaea invest in carbon-acquisition pathways to broaden their resource usage. These findings suggest divergent directions in eco-evolutionary adaptations to soil saline stress amongst microbial clades and serve as a foundation for understanding the response of soil microbiomes to escalating climate change.

Soil salinization is one of the most challenging global environmental concerns in the current century due to its negative impact on soil fertility and food security[1]. In the context of climate change, rising sea levels, increased soil porewater evaporation, and groundwater depletion are expanding the extent of salinized soils by 1.0–2.0 Mha per year[2]. By 2050, 50% of arable land is expected to be severely affected by salinization[3]. The high osmotic stress combined with ion toxicity ($Na^+$, $Cl^-$ etc.) induced by soil salinization endangers the survival of soil microorganisms and hinders their ecological functions[4]. Evaluating the eco-evolutionary strategies that microorganisms employ in response to saline stress is therefore essential to understanding and predicting climate-driven impacts on terrestrial ecosystem functions.

In general, microorganisms acclimate to stress by optimizing their genomic content in order to take advantage of specific ecological niches. This occurs through a series of resistance mechanisms including activation of the ion pump system[5,6] and accumulation of organic solutes to maintain osmotic pressure[7,8]. However, these strategies come at substantial carbon (C) and energy investment costs[9–12], which can result in intracellular C reallocation and reduced growth rates[13]. Thus, C limitation also influences microbial growth and behavior in extreme environments. This tradeoff between stress resistance and C limitation is a fundamental driver of eco-evolution in response to salinity and other stressors.

[1]College of Chemical Engineering, Nanjing Forestry University, Nanjing 210037, China. [2]Marine Agriculture Research Center, Tobacco Research Institute, Chinese Academy of Agricultural Sciences, Qingdao 266101, China. [3]State Key Laboratory of Soil and Sustainable Agriculture, Institute of Soil Science, Chinese Academy of Sciences, Nanjing 210008, China. [4]Earth and Biological Sciences Directorate, Pacific Northwest National Laboratory, P.O. Box 999 Richland, WA 99352, USA. [5]School of Biological Sciences, Washington State University, P.O. Box 645910 Pullman, WA 99164, USA. [6]Jiangsu Collaborative Innovation Center for Solid Organic Waste Resource Utilization, Nanjing 210095, China. ✉e-mail: rrchen@njfu.edu.cn; emily.graham@pnnl.gov

Variation in genome size is often coupled to microbial strategies for mitigating C limitation. The most commonly invoked eco-evolutionary strategy in microorganisms is the Black Queen Hypothesis (BQH), or so-called streamlining theory, which states that microorganisms reduce their genome size in response to energy constraints created by environmental stress[14–16]. This is because small genomes are directly linked with low energy costs for encoding and expressing fewer genes[17,18]. Additionally, small genomes are associated with smaller cell sizes, which can optimize the surface-to-volume ratio for the uptake of scarce nutrients[14,15]. All of these traits may confer fitness advantages in reducing energy expenditure[19], but in parallel, the reduced functional potential that can result in lower adaptability to environmental fluctuations[20]. The vast majority of microorganisms thriving in environments with high temperatures[21–23] and low pH[19,24,25] have been reported to feature small and streamlined genomes.

The Red Queen Hypothesis (RQH), in contrast, posits that species continuously evolve to maintain fitness by increasing their functional complexity (i.e. gaining genes)[26]. This hints that microorganisms could invest in multiple energy acquisition mechanisms to enhance their ability to acquire C, and thus cope with environmental pressure while maintaining a distribution of functions. Until now, the application of RQH has mostly occurred in the context of eukaryotic co-evolution (e.g., host-parasite interactions), and it has not been directly reported as a stress adaption strategy for microorganisms. However, insights from marine microorganisms suggest that the RQH may apply to some prokaryote microorganisms. Most marine microorganisms have small genomes (< 1–2 Mb)[27,28], yet some marine microorganisms have unusually large genomes[16], suggesting an alternative evolutionary strategy to genome streamlining.

Thus, we hypothesize that soil microorganisms predominately adapt to high salt concentrations through genome streamlining, but that some clades are able to survive in highly saline environments while maintaining an array of biogeochemical functions. If the latter proves to be true, a natural follow-on question is which microbial clades are more likely to adapt to stress in accordance with the RQH. Archaea have significantly smaller genome sizes than bacteria—over 90% of archaeal genomes are <2 Mb in size, while only 25% of bacterial genomes fall within this size range[29]. Considering that the extremely compact genome size in archaea makes further genome size reductions unlikely, we further hypothesize that archaea are likely to employ alternative eco-evolutionary strategies like the RQH.

At the community level, assessing microorganisms through a trait-based framework helps us understand the feedback of microbes to the environments, which may not be detectable through taxonomic analyses alone. Genomic traits such as genome size and number of regulatory genes are relatively easy to obtain, making them ideal metrics for large-scale comparisons and potentially valuable tools for linking microbial communities with ecosystem-level processes[30].

In this work, to test the above hypotheses, we collect soils along a salinity gradient and then combine shotgun metagenomic and amplicon sequencing to distinguish bacterial and archaeal clades with different responses to salinity. We then compare genome sizes and functional traits including Kyoto Encyclopedia of Genes and Genomes (KEGG) pathways, overall KEGG Orthologies (KOs), and specific KOs/genes encoding for salt-resistance and carbon-acquisition (C-acquisition). We illustrate that archaea explore novel sources of resources while bacteria optimize the use of existing resources under salt stress. Our results reveal eco-evolutionary strategies of bacteria and archaea for relieving C limitation evolve in two different directions.

## Results

### Response of microbial community structure and potential function to salinity

Soil salinization appeared to drive changes in prokaryotic community diversity. 16 S rRNA amplicon sequencing showed a significant decrease in alpha diversity with increasing soil salinity (OLS regression, $P < 0.05$, Fig. S3a) and a clear variation in beta diversity (Fig. S3b, c, Fig. S4). This succession of microbial communities to salinity is mainly driven by taxon-specific differences in salt tolerance.

To investigate possible eco-evolutionary strategies of salt-tolerant bacteria and archaea, we screened 500 taxa with positive and negative responses to salinity (see Method section for details), which are dispersed in bacterial and archaeal clades. These 500 taxa formed four observation groups as their differences in clades and response to salinity. Specifically, they were bacteria with a negative response to salinity (neg-bac, 200 taxa), bacteria with a positive response to salinity (pos-bac, 200 taxa), archaea with a negative response to salinity (neg-arch, 50 taxa), and archaea with a positive response to salinity (pos-arch, 50 taxa), respectively, for a total of 500 taxa (Fig. S2). These 500 taxa accounted for about 50% abundance of total taxa in the vast majority of samples (Fig. 1a), and their distribution at the phylum level is extremely heterogeneous (Fig. 2a).

We found that bacteria and archaea behave differently in salinity. Taxa in pos-bac and neg-bac tended to be linearly associated with salinity (Fig. 1b, c), while taxa in pos-arch and neg-arch exhibited thresholding behavior characterized by a breakpoint in the relationship between absolute abundance and salinity (Fig. 1d, e). A threshold value of EC was observed at $3.4 \, dS \, m^{-1}$ in the neg-arch group, and $4.2 \, dS \, m^{-1}$ in pos-arch group.

The functional potential of soil microbiomes changed synergistically with community structure in response to salt stress (Fig. S5). Based on metagenomic sequencing, the richness of overall KOs (Fig. S5a) and the abundance of overall KOs carried by taxon (Fig. S5c) were negatively related to salinity ($P < 0.01$), but the corresponding Shannon index was positively correlated to salinity (Fig. S5b, $P < 0.001$). Additionally, the β-diversity of overall KOs associated with the four salt response groups showed obvious differences (Fig. S7).

### Patterns in microbial genome size in response to salinity

We observed contrasting responses of bacterial and archaeal genome size to salinity. For bacteria, the positive response group had significantly smaller genome sizes on average than the negative response group, while archaeal taxa with positive responses to salinity had significantly larger genome sizes than those with negative responses (Fig. 2b, $P < 0.001$). Additionally, the difference in genome size between the archaeal response groups was greater than that between the two bacterial response groups.

To rule out phylogenetic differences in genome size that may confound these results, we further compared the genome size of taxa in the same phylum that belonged to positive vs. negative response groups. For bacteria, only *Proteobacteria* and *Chloroflexi* were shared in the neg-bac and pos-bac groups. Considering the low abundance of *Chloroflexi* in two response groups (<2%, Fig. 2a and Fig. S6), as well as its small taxa number (10 taxa vs 6 taxa) is not sufficient to support the reliability of the statistical analysis. Thus, only *Proteobacteria* (112 taxa vs 69 taxa) were further analyzed. *Proteobacteria* that responded positively to salinity had significantly smaller genome sizes than those that responded negatively (Fig. 2c, $P < 0.001$). There were no common phyla between neg-arch and pos-arch, as all taxa found in pos-arch belonged to *Euryarchaeota*, and all taxa in neg-arch belonged to *Thaumarchaeota* (Fig. 2a and Fig. S6). Therefore, the genome size distribution for each archaeal response group was compared to all members of *Euryarchaeota* (pos) or *Thaumarchaeota* (neg) regardless of their response to salinity. Taxa in pos-arch had an average genome size of 3.74 Mb, which was significantly larger than the whole *Euryarchaeota* phylum with 2.49 Mb (Fig. 2d, $P < 0.001$). A survey-analysis was further conducted to show the genome size of previously reported salt-tolerant archaea. Notably, these surveyed salt-tolerant archaea all belong to *Euryarchaeota* (Supplementary Data. 3). The average genome size of surveyed salt-tolerant archaea was 3.53 Mb, which was

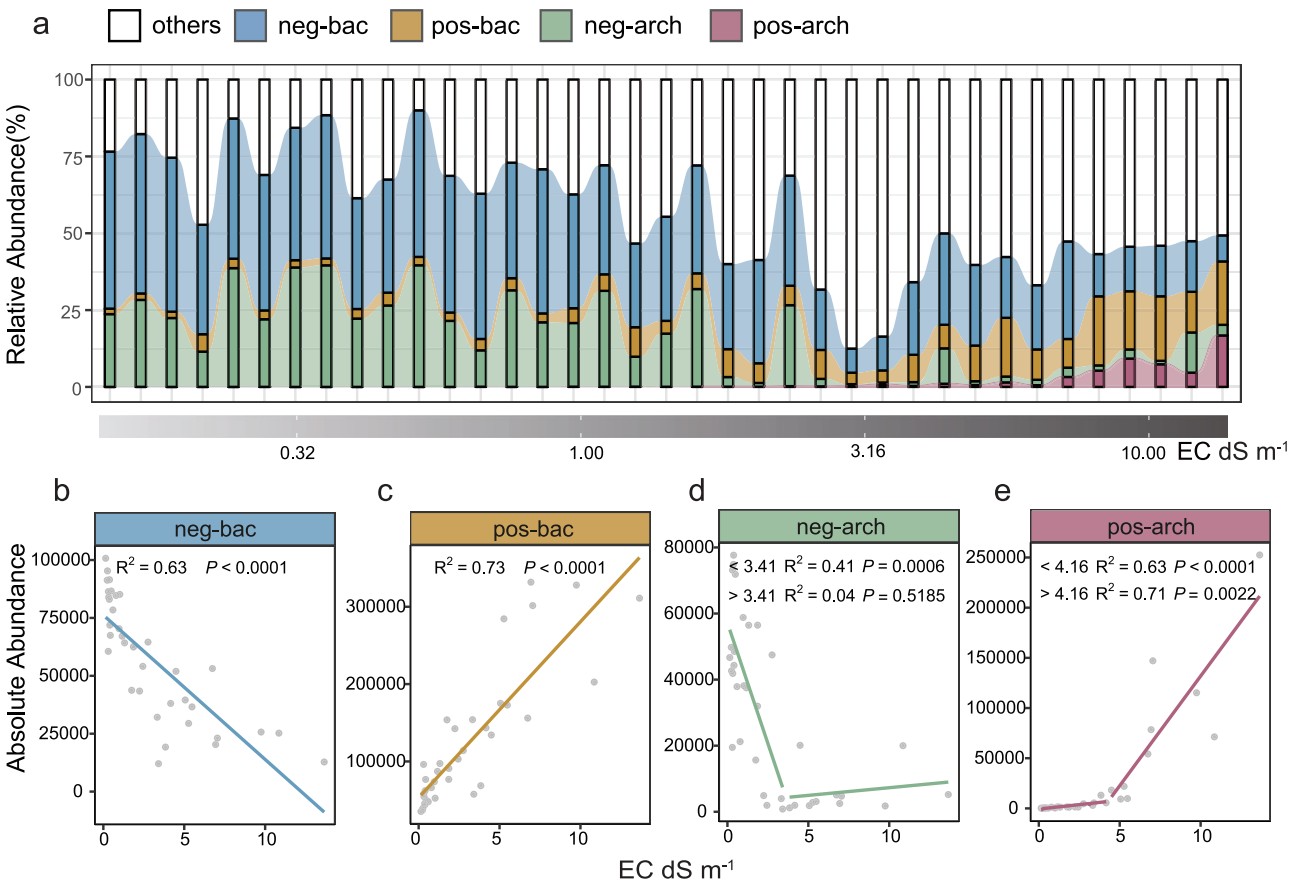

**Fig. 1 | Taxonomic succession along the soil salinity gradient. a** Relative abundance of each response group. **b−e** Absolute abundance of each response group. *P*-values are calculated by two-sided ordinary least squares linear regression (*P* < 0.05 indicates a significant correlation). Abbreviation: EC, electrical conductivity. Source data are provided as a Source Data file.

close to the pos-arch group (3.74 Mb) in the current study and significantly larger than that of the whole *Euryarchaeota* (Fig. 2d, *P* < 0.001). Although neg-arch had a larger genome size than *Thaumarchaeota* holistically, the difference between them (1.60 Mb vs. 1.36 Mb) was less dramatic than that between pos-arch and *Euryarchaeota* (3.74 Mb vs. 2.49 Mb).

**Response of microbial functional potential to salinity**

In both bacteria and archaea, smaller genome size tended to be associated with a lower abundance of functional genes. On average, taxa in the pos-arch group had a significantly higher abundance of KOs and KEGG pathways than taxa in neg-arch, while the opposite was observed in bacteria (Fig. 3a, b, *P* < 0.001). There was only one exception, genetic information processing pathways had no significant difference between the pos-bac and neg-bac groups (Fig. 3b, *P* > 0.05).

We further determined level 3 KEGG pathways that distinguished salt-tolerant groups from salt-sensitive groups, resulting in 39 bacterial and 51 archaeal pathways (Fig. 3c). For bacteria, 31 of 39 biomarker pathways were metabolism related, and 29 of these 31 metabolism-related pathways were more abundant in the neg-bac group than in the pos-bac group. Conversely, among the 51 archaeal biomarker pathways, 35 were related to metabolism, but 31 of these 35 were more abundant in the pos-arch group than in the neg-arch group. Specific to genetic information processing pathways, RNA polymerase was more abundant in the neg-bac group than in pos-bac, but homologous recombination showed the opposite pattern. The neg-arch had a higher abundance for 3 of 4 genetic information processing pathways than the pos-arch, including protein processing in endoplasmic reticulum, base excision repair, and non-homologous end-joining.

We also found that bacteria and archaea had divergent genomic responses to salinity. As the salinity increased, the overall abundance of KOs in bacterial communities decreased (Fig. S8a, *P* < 0.05), while that of archaea increased (Fig. S8b, *P* < 0.05). Then, the KEGG pathways that are susceptible to salinity in four response groups were separately explored by random forest analysis. The pathways including mismatch repair, DNA replication, fatty acid metabolism, fatty acid degradation, and glycine, serine and threonine metabolism, and cofactor/vitamin metabolism were significantly enriched in the pos-arch group by salt (Fig. S8c).

**Salt-resistance genes in salt-tolerant bacteria and archaea**

Both salt-tolerant bacteria and archaea had a significantly higher abundance of KOs associated with salt-resistance compared to salt-sensitive taxa (Fig. 4a, *P* < 0.001). This provides support for the accuracy of our grouping. Different from the abundance, substantial differences in the diversity of salt-resistance mechanisms were observed between bacteria and archaea. Salt-tolerant archaea tended to possess comprehensive salt-resistance mechanisms. For the three most well-known adaptive salt-resistance mechanisms (including Na$^+$ extrusion, K$^+$ uptake and synthesizing osmotic solute, more details see Methods and Supplementary Data. 5), 76% of tolerant archaea (38 taxa) possessed all three resistance mechanisms (Fig. 4e), while only 20% of sensitive archaea (10 taxa) had all three (Fig. 4d). The opposite pattern was shown by the bacterial groups. That is, taxa in pos-bac had lower diversity of salt-resistance mechanisms.

**C-acquisition potential in salt-tolerant bacteria and archaea**

Salt-tolerant bacteria and archaea tended to contain a higher proportion of C-acquisition genes (relative gene abundance, the proportion

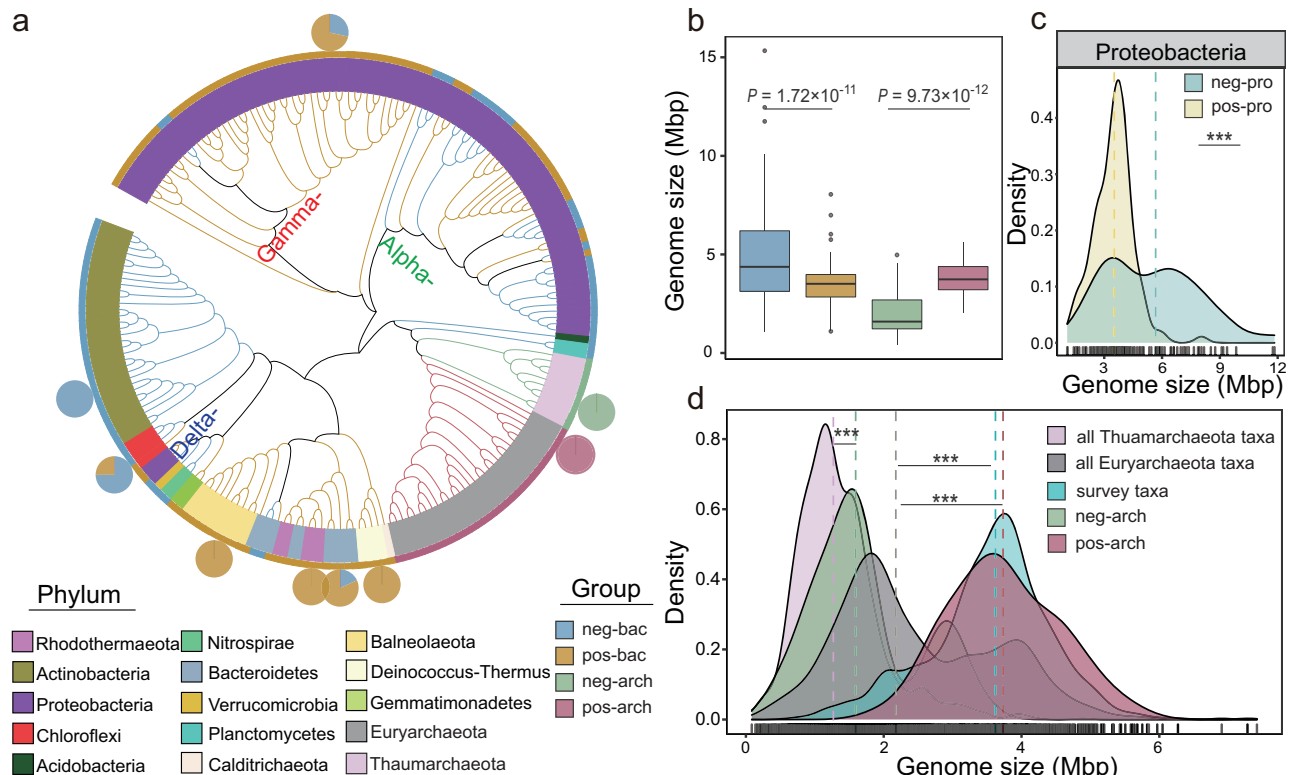

**Fig. 2 | Phylogenetic distribution and genome size for each response group.**
**a** Phylogenetic distribution. **b** Genome size comparison of bacterial/archaeal response groups ($n = 194$ for neg-bac group; $n = 196$ for pos-bac group; $n = 44$ for neg-arch group; $n = 47$ for pos-arch group). **c** Genome size comparison within Proteobacteria in two bacterial response groups ($n = 69$ for neg-bac group; $n = 112$ for pos-bac group). **d** Frequency distribution and comparison of taxa genome size in archaea positively and negatively responding to salinity, of all *Euryarchaeota* and *Thaumarchaeota*, and of salt-tolerant archaea found in published literature ($n = 1369$ for all Thaumarchaeota taxa group; $n = 5389$ for all Euryarchaeota taxa

group; $n = 1603$ for survey taxa group; $n = 44$ for neg-arch group; $n = 47$ for pos-arch group). The pie chart in the phylogenetic tree represents the distribution of taxa of adjacent phyla in the four response groups. Boxplots indicate the median (middle line), 25th, 75th percentile (box) and 5th and 95th percentile (whiskers). Differences in (**b**, **c** and **d**) were tested using unpaired two-sided Mann–Whitney $U$-test with 95% confidence interval. The dashed lines in (**c**) and (**d**) denote the median of the dataset. The asterisk ***$P < 0.001$ (detailed $P$-values are shown in source data). Source data are provided as a Source Data file.

of C-acquisition gene abundance to overall gene abundance) than the salt-sensitive counterparts (Fig. 5a). Specifically, the relative abundance of total C-acquisition genes (carbon-fixation (C-fixation)) genes plus carbon-degradation (carbohydrate-active enzymes, CAZymes) in the pos-bac group was 13.49% vs 12.43% in neg-bac. This pattern was more obvious for archaea, with a total C-acquisition genes ratio of 24.37% in pos-arch vs. 12.65% in neg-arch. There was no significant difference in the absolute abundance of genes involved in C-acquisition between pos-bac and neg-bac (Fig. 5b, $P > 0.05$). For archaea, pos-arch had a significantly higher absolute abundance of genes involved in C-acquisition than neg-arch (Fig. 5b, $P < 0.001$).

We also found variation in C-acquisition gene diversity between bacterial and archaeal response groups (Fig. 5c, d, Fig. S9, and Table S1). Frequency distributions of C-acquisition genes indicated that salt-tolerant archaea tended to have more comprehensive C-acquisition mechanisms than salt-sensitive archaea, while the salt-tolerant bacteria contained the lower diversity of C-acquisition mechanisms relative to salt-sensitive bacteria. Thirty-four taxa in the pos-arch group (68%) had all seven C-fixation mechanisms (the C-fixation mechanisms were listed in Supplementary Data. 4), but only 11 taxa with all seven mechanisms (22%) were found in neg-arch. In comparison, only 69 taxa in pos-bac (34.5%) had all seven C-fixation mechanisms assayed, lower than 163 in neg-bac (81.5%). All taxa in the pos-arch group had five or more C-fixation mechanisms, while there were 22 taxa in neg-arch (44%) without inorganic C-fixation mechanisms (Fig. 5c). These patterns also applied to C-degradation CAZymes. In short, there were 35 taxa with all six categories of CAZymes in the

pos-arch group (accounting for 70%), versus 6 (12%) in neg-arch (Fig. 5d). Bacteria had the opposite pattern, with 86 taxa in pos-bac (43%) and 139 (69.5%) in neg-bac (Fig. 5d).

## Discussion
Salt, as a strong environmental filter on microorganisms, significantly structured microbial communities[31,32], and seemed to result in divergent eco-evolutionary strategies in bacterial vs. archaeal microorganisms. Selection appeared to be particularly strong in archaea–all of the 50 salt-tolerant archaea belonged to the *Euryarchaeota*, while almost all salt-sensitive archaea were *Thaumarchaeota* (Fig. 2a). We hypothesized that these changes in soil microbial community structure would be predominately related to genome streamlining, but that some microbial clades would be able to maintain a stable genome size and diverse suite of potential functions in response to stress. We further hypothesized that archaeal communities may be less likely to experience genome streamlining due to their smaller average genome size than bacteria. As soil salinity increases with climate changes, understanding microbial adaptations to salt stress is essential for predicting future biogeochemical functions in soils.

### Reduction in bacterial genome size in highly saline soils
Changes in bacterial genome size with salinity were consistent with genome streamlining as an eco-evolutionary strategy. Gene loss was collectively demonstrated by the smaller genome size of salt-tolerant bacteria, as well as the less functional genes of the bacterial taxa with salinity gradient (Fig. 3a and Fig. S8a). We propose that this may be a

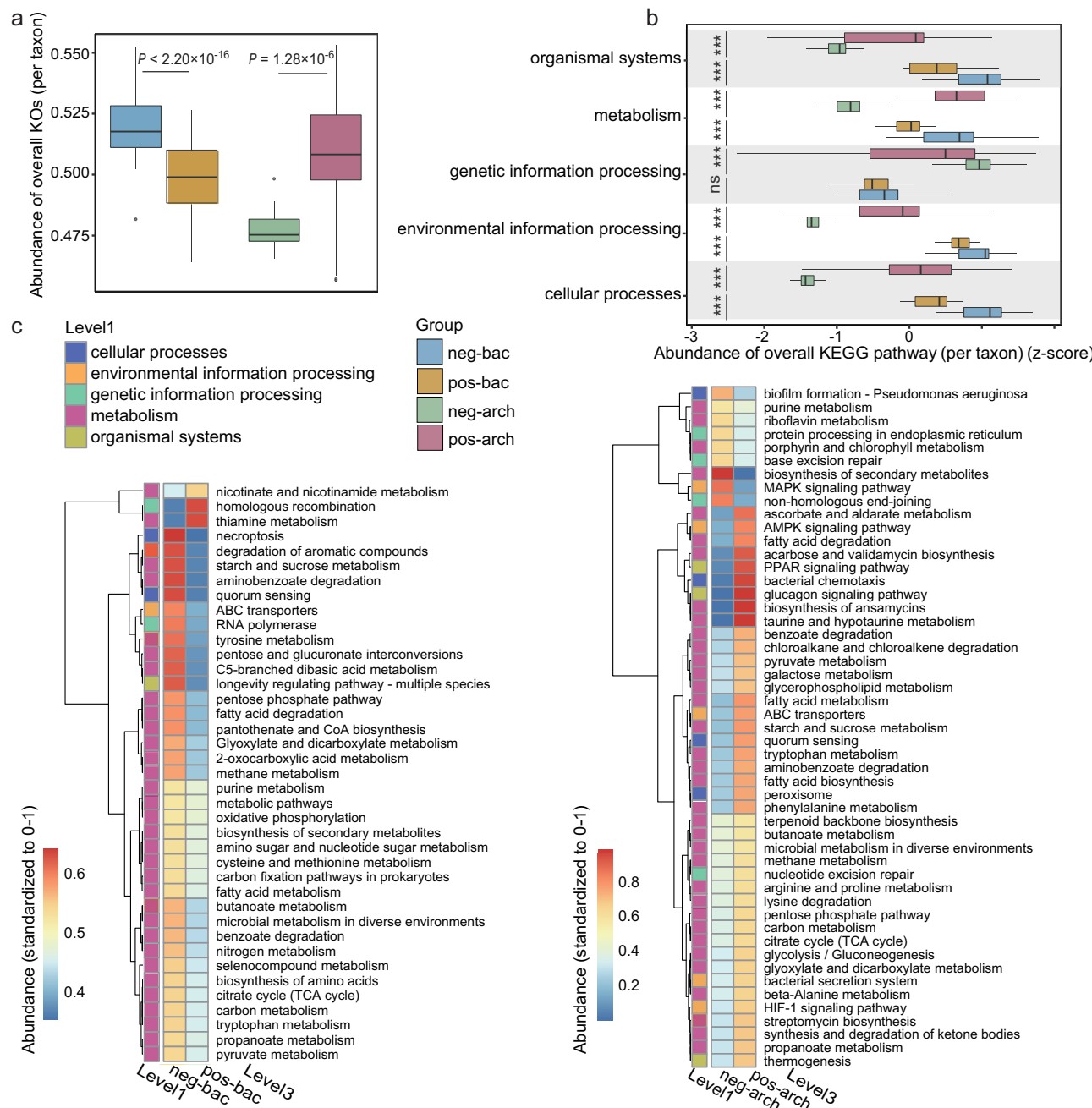

**Fig. 3 | Changes in bacterial and archaeal functional potential in response to salinity. a** Total abundance of KOs per taxon in each response group ($n = 200$ for neg-bac and pos-bac groups; $n = 50$ for neg-arch and pos-arch groups). **b** Abundance of KEGG pathways (normalized by taxa abundance) at BRITE hierarchy level 1 per taxon in each response group (z-score). **c** Heatmap of total abundance of differential KEGG pathways at BRITE hierarchy level 3 in each

response group. Boxplots indicate the median (middle line), 25th, 75th percentile (box) and 5th and 95th percentile (whiskers). Differences in (**a**) and (**b**) were tested using unpaired two-sided Mann–Whitney U-test with 95% confidence interval. The asterisk ***$P < 0.001$, and ns denotes $P > 0.05$ (detailed P-values are shown in source data). The whiskers in (**c**) represent the clustering of KEGG pathways. Source data are provided as a Source Data file.

common bacterial eco-evolutionary response to stress. For example, a meta-analysis based on the genome size of 260 acidophilic bacteria reported that the genome size of acidophilus was significantly smaller than that of the closest lineages living in circum-neutral pH environments[19]. The deletion of intergenic regions has been reported to contribute to the reduction in genome size[33], which could not be ascertained from our data. However, in this study, the changes in the overall abundance of KOs were consistent with the genome size, strongly suggesting a pattern of protein-encoding gene loss leading to genome reduction (Fig. 3a)[34].

Gene loss appeared to be selective in bacteria, as >60% KEGG pathways that experienced gene losses were related to metabolism (Fig. 3c). These diluted metabolism pathways included carbohydrate metabolism, amino acid metabolism, and cofactor/vitamin metabolism (Fig. 3c and Fig. S8c), which are closely related to bacterial growth. This result suggested a tradeoff between adaptations to extreme environments at the expense of growth[35,36]. Functionally redundant genes were also disproportionally lost during genome streamlining[37]. In this study, salt-tolerant bacteria with small genomes contained a lower diversity of C-acquisition and salt-resistance mechanisms than

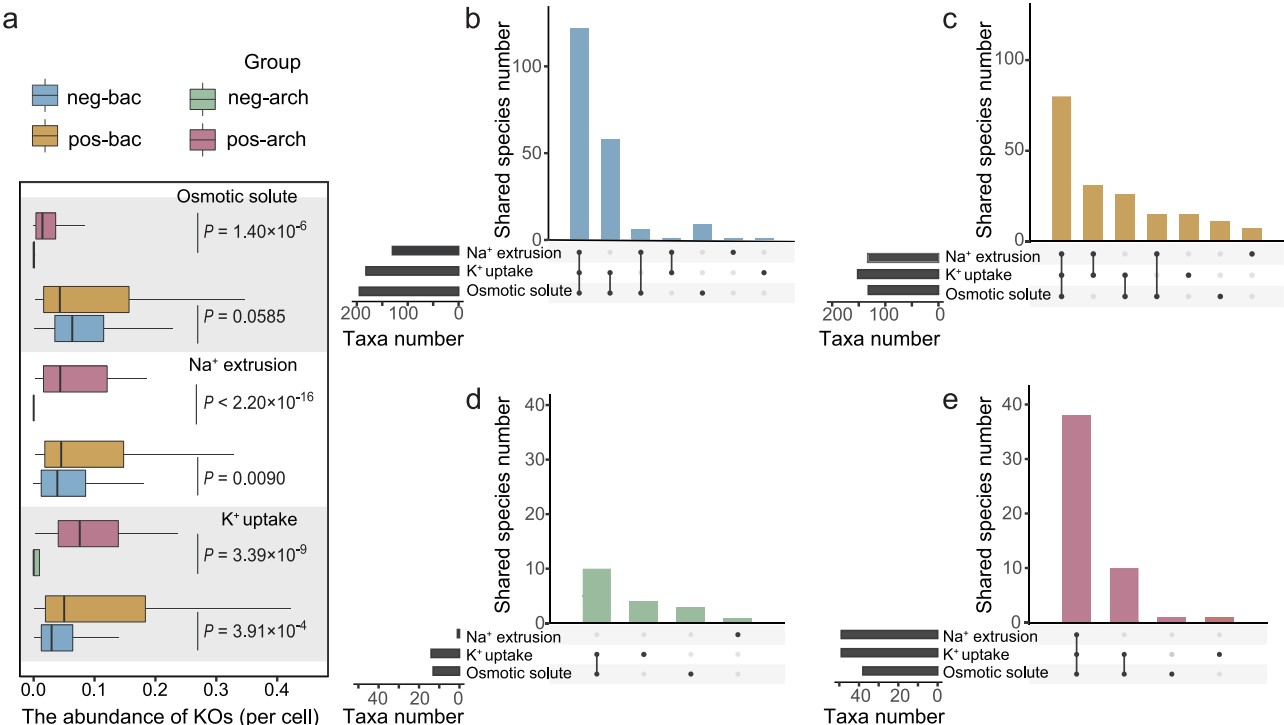

**Fig. 4 | Differences in salt-resistance genes among bacterial and archaeal salinity response groups. a** Abundance of KOs associated with salt-resistance (normalized by taxa abundance, $n = 200$ for neg-bac and pos-bac groups; $n = 50$ for neg-arch and pos-arch groups). **b–e** Distribution frequencies of taxa containing each salt-resistance mechanism in each response group visualized with UpSet plots. Boxplots indicate the median (middle line), 25th, 75th percentile (box) and 5th and 95th percentile (whiskers). Differences in (**a**) were tested using unpaired two-sided Mann–Whitney $U$-test with 95% confidence interval. The black bars denote the counts of taxa with the corresponding mechanism. The black dots denote the number of taxa with unique (one dot) or shared (≥two dots) mechanism(s) in each group. Source data are provided as a Source Data file.

salt-sensitive bacteria (Figs. 4b, c, 5c and d). This is consistent with previous work reporting that environmental stress such as low pH, hyperthermia, drought, and salt caused the loss of functionally redundant genes[38]. In particular, the salt-tolerant bacteria were observed to have a higher tendency to sacrifice their salt-resistance mechanisms in synthesizing organic osmotic solutes (Fig. 4b, c). This is because synthesis of organic solutes is a very energy expensive stress mechanism, which can cause a reduction of growth yields (i.e., biomass produced per gram C metabolized) by roughly 90%[35].

Despite genome streamlining as an overarching strategy in salt-tolerant bacteria, we found that some genes were robust or enriched in salt-tolerant bacteria. These included genes associated with genetic information processing (Fig. 3b), which are usually considered core/housekeeping genes encoding basic cellular activities, e.g., replication, transcription, and translation[34,39,40]. Additionally, genes for salt-resistance were enriched in salt-tolerant bacteria (e.g., $Na^+$ extrusion, $K^+$ uptake, organic solutes absorption or synthesis, Fig. 4a), which can maintain osmotic pressure and cellular physiological metabolism within desired limits[41,42]. Also enriched were KEGG pathways of glycine, serine and threonine metabolism (Fig. S8c), which were involved in the synthesis of osmotic solutes betaine and ectoine[43].

### Increase in archaeal genome size in highly saline soils

Interestingly, our results showed salt-tolerant archaea had significantly enlarged genome size compared to salt-sensitive archaea (Fig. 2), suggesting an eco-evolutionary strategy that may be completely divergent from bacteria. The average genome size of salt-tolerant archaea in the current study and literature were both significantly higher than the *Euryarchaeota* phylum as a whole (Fig. 2d). To our knowledge, this is the first report of archaeal genome enlargement under environmental stress.

The expanded genome of salt-tolerant archaea was accompanied by increased functional trait (Fig. 3a). While an observed increase in salt-resistance genes was expected (Fig. 4), we also identified enriched genes across a wide distribution of KEGG pathways (Fig. 3b, c). Gene enrichment was especially concentrated amongst genes related to cellular metabolism. At BRITE hierarchy level 3, KEGG pathways including carbohydrate metabolism, amino acid metabolism, and lipid metabolism were enriched in salt-tolerant archaea (Fig. 3c). Moreover, the abundance of these metabolic pathways in salt-tolerant archaea, as well as DNA replication, significantly increased along the salinity gradient (Fig. S8c). Notably, these genes in salt-tolerant bacteria are much lower than those in salt-sensitive bacteria (Fig. S8c), supporting divergent adaptations to salinity between archaea and bacteria. Collectively, the enriched metabolic genes in salt-tolerant archaea suggest that salinity may not hinder their growth, in sharp contrast to the genomic strategies of salt-tolerant bacteria.

Genomic adaptation of salt-tolerant archaea may explain why salt-tolerant archaea seemed to outcompete salt-tolerant bacteria in highly saline soils. Salt-tolerant archaea tended to have more comprehensive metabolic mechanisms than salt-tolerant bacteria (Fig. 5c, d), which may confer an advantage in resource acquisition and population proliferation. An experiment on antibiotic susceptibility supported that fast growth can counteract antibiotic stress in driving community transitions[44]. Secondly, in contrast to genome streamlining, archaea with large genomes appeared to contain functional redundancy, implying that they are less dependent on interspecies interactions for survival[45]. Frequent interactions between species may be detrimental to community stability in some cases[46], as the break of any link in the interaction may lead to the collapse of the entire community. The preservation of genome size and functional redundancy by salt-tolerant archaea, as an alternative evolutionary strategy with

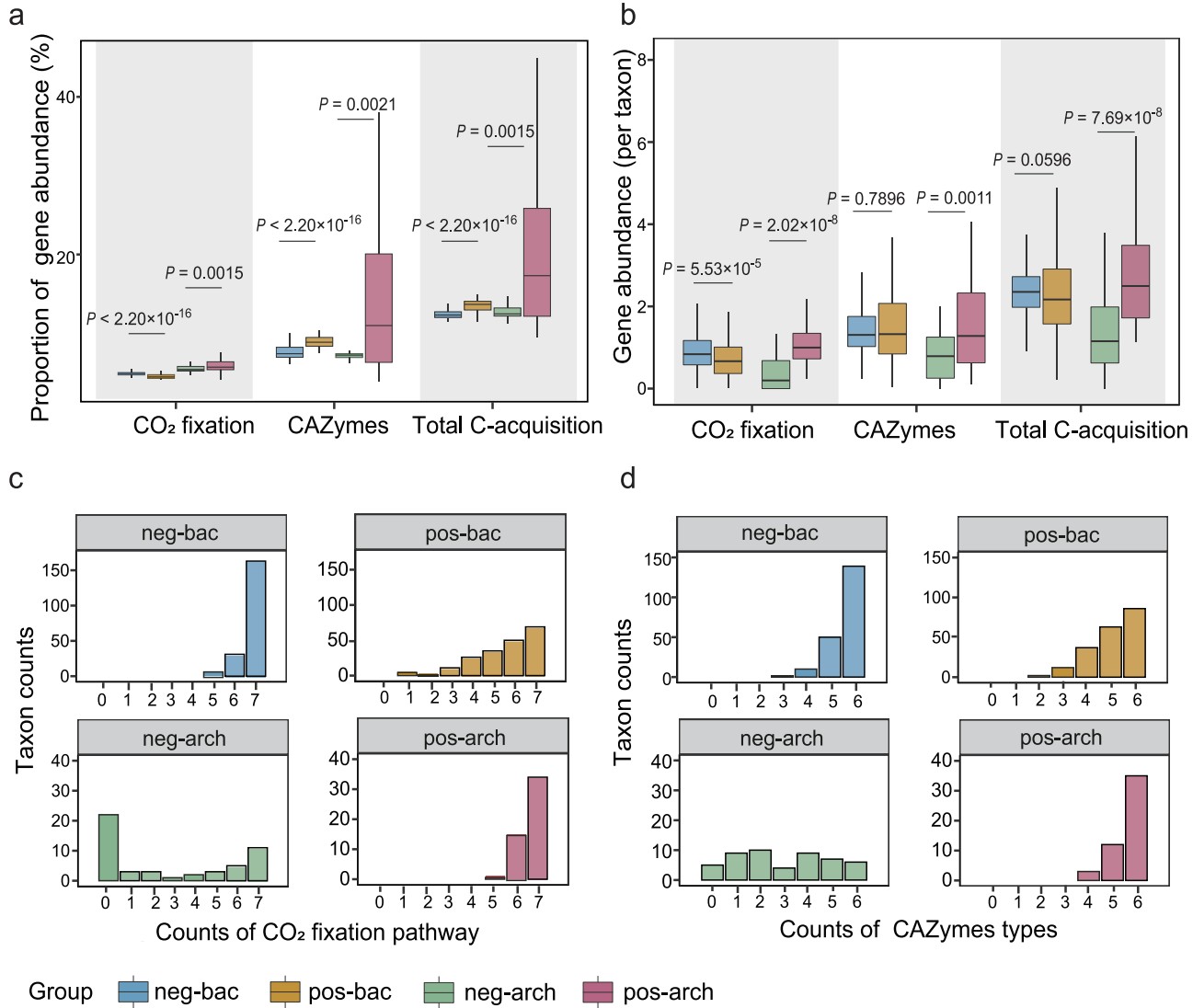

**Fig. 5 | Differences in bacterial and archaeal C-acquisition genes in response to salinity. a** Proportion of C-acquisition genes relative to overall gene abundance. **b** Abundance of genes associated with C-acquisition (normalized by taxa abundance). **c** Frequency distribution of genes related to seven main mechanisms for C-fixation across taxa. **d** Frequency distribution of genes related to six types of CAZymes for C-degradation across taxa. Total C-acquisition is the sum of the abundances of $CO_2$ fixation gene and CAZymes gene. Boxplots indicate the median (middle line), 25th, 75th percentile (box) and 5th and 95th percentile (whiskers). Differences in (**a**) and (**b**) were tested using unpaired two-sided Mann–Whitney $U$-test with 95% confidence interval ($n = 200$ for neg-bac and pos-bac groups; $n = 50$ for neg-arch and pos-arch groups). Abbreviation: CAZymes, carbohydrate-active enzymes. Source data are provided as a Source Data file.

advantages over genome streamlining, holds significant importance in comprehending the adaptive capacity of microbial populations when confronted with environmental changes.

**C-acquisition strategies differ between salt-tolerant bacteria and archaea**

As a core element of microbial composition and energy supply, C plays an important role in microbial growth and stress resistance[35,36]. Salt-resistance mechanisms are also energy (C) intensive[9], especially the synthesis of osmotic solutes[35,47]. Our results showed that both salt-tolerant bacteria and salt-tolerant archaea contained a higher proportion of C-acquisition genes than their salt-sensitive counterparts (Fig. 5a), putatively indicating increased investment in C-acquisition under salt stress. This provided evidence that salt-tolerant taxa (regardless of bacteria or archaea) increased C demand, as they encoded more salt-resistance genes than salt-sensitive taxa (Fig. 4a).

Despite common increased demands for C under salt stress, distinct patterns in genome size and gene content between bacteria and archaea signify distinct eco-evolutionary directions associated with C-acquisition and usage. Reduced genome size (Fig. 2b) and functional gene content (Fig. 3a) in salt-tolerant bacteria are putatively associated with lower C and energy consumption. We found that salt-tolerant bacteria did not significantly enhance C-acquisition ability compared with salt-sensitive bacteria, as evidenced by absolute gene abundance of C-acquisition (Fig. 5b). Thus, we propose that salt-tolerant bacteria economize energy consumption to alleviate C limitation on growth by adopting a strategy consistent with the BQH (genome streamlining)[15]. Salt-tolerant archaea, in contrast, had the highest abundance of C-acquisition genes (Fig. 5b) and contained more comprehensive C-acquisition mechanisms overall (Fig. 5c and d). This potentially enables archaea to fulfill their C supply requirements, supporting both mechanisms of salt stress resistance and the maintenance of enlarged genome. Therefore, we propose that salt-tolerant archaea relieve C limitation by increasing investments in C-acquisition genes, consistent with the RQH and in direct contrast with bacterial lifestyles.

In summary, our findings showed contrasting evolutionary directions of soil bacteria and archaea in response to increases in salinity. Bacteria appear to employ a genome streamlining strategy, evidenced by small genomes with a decreased abundance of functional genes in response to salt. However, archaea maintain genome size and increase functional complexity under salt stress. In addition, genes related to C-acquisition were enriched in salt-tolerant archaea, suggesting that they maintained high C-acquisition potential despite saline stress. These results uncovered that salt-tolerant bacteria appear to economize energy consumption through genome streamlining, while salt-tolerant archaea broaden their resource usage by increasing C-acquisition capacity. Divergent evolutionary strategies may in turn change population interactions and community succession amongst bacteria and archaea. This finding broadens our understanding of eco-evolutionary adaptations to environmental stress and serves as a foundation for understanding the response of soil microbial communities to salinization with escalating climate change.

## Methods

### Site description and sample collection
The coastal soil samples were collected in Dongying (37°51′N-38°60′N, 118°19′E-118°58′E) near the border of the Bohai Gulf, north of Shandong Province, China. The climate in this area is monsoon continental, with an average annual temperature of 12.8°C and an average annual precipitation of 555.9 mm (obtained from http://www.weather.com.cn)[48]. Soil salinization in this region resulted from the combined effects of natural processes as well as anthropogenic activities such as landforms, climate, and land uses[49]. Based on the distance from the coastline and the type of covered vegetation, a total of 37 topsoil samples were collected (Fig. S1a and S1b). For each site, 20 cores of topsoil (0-20 cm) were extracted from a 10 * 10 (m$^2$) quadrat with a serpentine sampling method, using a 30 mm-diameter gouge auger. Notably, the soils studied were non-rhizosphere soils, so vegetated areas should be avoided when collecting samples. Each sample was homogenized and sieved (< 2 mm) prior to splitting into two subsamples. EC was determined using an electronic conductivity meter (Mettler Toledo, OH, USA) for suspended soils at a soil-water ratio of 1:5 (w/v). There was a significantly negative correlation between EC and the distance of the sampling site from the coastline ($P < 0.01$, Fig. S1c). And according to the EC of the dried soils (ranging from 0.14 dS m$^{-1}$ to 13.65 dS m$^{-1}$), the salinity gradient of these soil samples was formed. Subsamples used for DNA extraction were stored at −40°C, and the others were air-dried for chemical analysis. The detailed site description and edaphic properties from the 37 samples were supplied in Supplementary Data. 1.

### DNA extraction
Genomic DNA was extracted from 0.5 fresh soil with a FastDNA® SPIN Kit for soil (MP Biomedicals, Santa Ana, CA). The extracted DNA was dissolved in 50 μL TE buffer and quantified by NanoDrop ND−2000 (Thermo Fisher, Waltham, MA, USA) and stored at −40°C until further use.

### High−throughput sequencing of 16 S rRNA gene fragments
PCR amplification was conducted with the universal primer set 519 F/907R[50], which amplified ~400 bp of 16 S rRNA gene V4–V5 fragments (519 F: 3′-CAGCMGCCGCGGTAATWC-5′; 907 R: 3′-CCGTCAATTCMTTTRAGTTT-5′). The unique oligonucleotides of 5 bp bar-coded were fused to the forward primer to distinguish different samples. PCR was carried out in 50 μL reaction mixture, containing deoxynucleotide triphosphate at a concentration of 1.25 μM, 2 μM of Taq DNA polymerase (TaKaRa, Japan), 2 μL (15 μM) forward and reverse primers, and each reaction mixture received 1 μL (50 ng) of genomic community DNA as a template. Amplification was conducted with the following program: 94°C for 5 min, 30 cycles (94°C for 30 s, 55°C for 30 s, 72°C

for 45 s), and a final extension at 72°C for 10 min. Reaction products for each sample were cleaned by the QIAquick PCR Purification Kit (Qiagen, Valencia, CA, USA), and quantified by NanoDrop ND−2000 (Thermo Scientific, Waltham, MA, USA). The purified bar-coded PCR products were pooled in equimolar amounts. High-throughput sequencing was performed with the Illumina MiSeq sequencing platform (Illumina Inc., CA, USA).

After sequencing, 16 S rRNA gene sequences were processed using the Quantitative Insights Into Microbial Ecology (QIIME1) pipeline for data sets (http://qiime.org/index-qiime1.html)[51]. Sequences with a quality score below 25 and a length of fewer than 200 bp were trimmed and then assigned to soil samples based on unique barcodes. A total of 2,301,297 high quality sequences were finally obtained (max = 77,558, min = 47,170, median = 61,389). The remaining sequences were further binned into operational taxonomic units (OTUs) with a 97% identity threshold, and the most abundant sequence from each OTU was selected as a representative sequence. Taxonomy was assigned to OTUs with reference to a subset of the SILVA 132 database (http://www.arb-silva.de/download/archive/qiime/)[52]. All samples were rarefied to 47,170 sequences to evaluate alpha- and beta-diversities of soil bacterial phylotypes.

### Shotgun metagenomic sequencing
The extracted DNA was fragmented to ~400 bp with Covaris M220 (Gene Company Limited, China). Then, the fragmented DNA was used to construct a paired-end library via TruSeq™ DNA Sample Prep Kit (Illumina, CA, USA)[53]. Shotgun metagenomic sequencing was performed on an Illumina HiSeq4000 platform (Illumina Inc., CA, USA) at Majorbio Bio-Pharm Technology Co., Ltd. (Shanghai, China). Sequencing produced ~10 Gbp paired-end Illumina data for each sample and the reads with low quality (with average quality scores <20 and length <50 bp) were removed (1.14–4.30% of all reads)[54].

The de Bruijn graph-based assembler SOAPdenovo (https://help.rc.ufl.edu/doc/SOAPdenovo, Version 1.06)[55] was employed to assemble short reads (Kmers range 47–97, step-10). K-mers varying from 1/3 to 2/3 of read lengths were then tested for all samples. Scaffolds with a length >500 bp were retained for statistical tests; the quality and quantity of scaffolds generated were evaluated by each assembly and chose the best Kmer, which yielded the maximum value of N50 and N90 and the minimum scaffold number, respectively. Scaffolds with a length >500 bp were then extracted and broken into contigs without gaps. These contigs were used for further gene prediction and annotation.

MetaGene was used to predict open reading frames (ORFs) from each metagenomics sample[56]. To obtain a non-redundant gene set, a pairwise comparison of predicted ORFs (filtered with a length being or over 100 bp) was performed using CD-HIT at 95% identity and 90% coverage[57].

Reads after quality control were mapped to the representative genes with 95% identity using SOAPaligner (https://help.rc.ufl.edu/doc/SOAPdenovo)[55]. BLASTP (Version 2.2.28 + , http://blast.ncbi.nlm.nih.gov/Blast.cgi) was employed for taxonomic annotation by aligning non-redundant gene catalogs against the NCBI NR database with an e-value cutoff of 1e − 5[58]. KEGG pathways annotation was conducted using the BLASTP search against the Kyoto Encyclopedia of Genes and Genomes database (http://www.genome.jp/kegg/.)[59] also with an e value cutoff of 1e − 5[53].

### Establish four response microbial groups with taxa responding inversely to salt
We established four responsive microbiomes as shown in Fig. S2. In brief, metagenome taxonomic annotations at the species-level resulted in 34,898 taxa. After removing taxa with a total abundance of <100 across all the samples, 15,718 taxa remained (14,949 bacteria and 769 archaea). The response of taxa to salinity was described by the linear

relationship between taxon abundance and EC. Consequently, 1896 taxa (1398 bacterial and 497 archaeal) responded positively to salinity ($P < 0.05$), and 5319 taxa (5248 bacterial and 71 archaeal) responded negatively to salinity ($P < 0.05$). Then, we ranked taxa in each response group by total abundance. The top 200 bacteria in each bacterial group, and the top 50 archaea in each archaeal group were selected for further investigation, which added up to a total of 500 taxa. All subsequent bioinformatic analyses were performed against the four response groups taking the abbreviation pos-bac (200 taxa), neg-bac (200 taxa), pos-arch (50 taxa), and neg-arch (50 taxa). Information on these 500 taxa is listed in Supplementary Data. 2.

We further evaluated the effectiveness of grouping to ensure accuracy, as well as avoid the interference of false positives on the results. The published salt-tolerant species were used as reference standards to observe the groups in which they were included. The published genera of salt-tolerant bacteria mainly include *Halo-*, *Salini-*, *Thio-*, *Ocean-*, *Marin-*, *Aquisalimonas*, *Aliifodinibius*, *Roseovarius*, *Rubrivirga*, *Vibrio*[60–62]. These genera were completely excluded from the neg-bac group. But they covered 81 taxa in the pos-bac group accounting for 61.8% of the identified genus-level taxa (69 taxa were not identified at the genus level). In addition, the remaining taxa in pos-bac group not included in the salt-tolerant genus were almost all extremophiles existing in Marine and alkaline lakes. The published class of salt-tolerant archaea mainly includes Halobacteria and Halophilic methanogenic archaea[63]. They were not included in the neg-arch group at all, but 100% covered the pos-arch group. The natural differences in salt tolerance between the response groups provided evidence that salinity is a key driver of genomic trait divergence. And it was further supported by the results of partial correlation between edaphic factors and abundance of overall KOs and C-acquisition genes, with EC as the control variable (Table S2).

## Data acquisition and processing

To assess changes in genome size in response to salinity, we obtained taxa genome sizes from the NCBI genome Database (https://www.ncbi.nlm.nih.gov/genome/)[64] for taxa in the four response taxa groups, as well as all taxa in the phyla Euryarchaeota (5390 taxa) and Thaumarchaeota (1369 taxa), and all halophilic archaea reported in a comprehensive literature search (1603 taxa). The genome size of the four response taxa is not complete, because very few taxa genome sizes are missing from the database (194 genome sizes observed for neg-bac; 196 genome sizes observed for pos-bac; 44 genome sizes observed for neg-arch; 47 genome sizes observed for pos-arch). The detailed data are shown in Supplementary Data. 3.

16 S ribosomal RNA gene sequences from 243 out of 500 taxa (ribosomal sequences for the remaining 257 species were not available) were downloaded from NCBI nucleotide Database (https://www.ncbi.nlm.nih.gov/nuccore/)[64]. PyNAST[51] was used to conduct multiple–sequence alignment of these ribosome sequences, and FastTree was used to construct the phylogenetic tree.

To assess changes in microbial traits in response to salinity, we focused on gene annotations from metagenomic sequencing that were associated with stress adaptations and C metabolism. We focused on the three dominant adaptive mechanisms for microorganisms to cope with salinity: (1) pumping $Na^+$ out of the cell through $Na^+/H^+$ antiporters to maintain the homeostasis of intracellular $Na^+$ [12,65], (2) accumulating $K^+$ via the $K^+$ transport system or $K^+/H^+$ antiporter[65], and (3) absorbing and/or synthesizing low molecular compounds to resist osmotic pressure[42]. Microbial KOs related to each of these mechanisms are listed in Supplementary Data. 5. Additionally, microorganisms acquire C mainly through inorganic carbon-fixation (C-fixation) and macromolecular organic carbon-degradation (C-degradation). There are seven inorganic carbon fixation pathways, including reductive pentose phosphate cycle (Calvin cycle), reductive citrate cycle (Arnon-Buchanan cycle), 3-Hydroxypropionate bi-cycle, hydroxypropionate-

hydroxybutylate cycle, dicarboxylate-hydroxybutyrate cycle, reductive acetyl-CoA pathway (Wood-Ljungdahl pathway), and incomplete reductive citrate cycle. The related KOs are listed in Supplementary Data. 4. And there are six categories CAZymes, including Auxiliary Activities (AAs), Carbohydrate Esterases (CEs), Carbohydrate-Binding Modules (CBMs), Glycoside Hydrolases (GHs), Polysaccharide Lyases (PLs), and GlycosylTransferases (GTs).

The abundance of KOs, KEGG pathways, specific KOs/genes encoding for salt-resistance, C-fixation, and carbohydrate-active enzymes (CAZymes) in each response group was composed of the abundance of all taxa within that group. Since our determination on taxa gene abundance is based on scaffolds, this means that their gene abundance is affected by their taxa abundance. To avoid this issue, we first normalized the abundance of functional genes by taxa abundance. *NACG* is the normalized abundance of the specific functional genes, which is calculated by dividing the abundance of a category genes within a taxon by that taxon abundance (Eq. (1)). In addition, the relative abundance of KOs related to C-fixation or CAZymes in the total functional gene abundances for each taxon was further calculated to estimate its investment in C (energy)-harvesting.

$$NACG_i = \frac{\sum_{j=1}^{n} ACG_{ij}}{AT_i} \tag{1}$$

in which, $NACG_i$ is the normalized abundance of each category gene in taxon $i$, (i.e., overall KOs, KEGG pathways, KOs involved in salt-resistance, and KOs/genes involved in C-acquisition). $ACG_{ij}$ represents the abundance of the gene $j$ of taxon $i$. $n$ is the number of genes that perform the corresponding functions. $AT_i$ is the abundance of taxon $i$.

## Statistical analyses

First, we explored variations in overall microbial community structure (OTUs based on 16 S rRNA gene sequences) and in KO composition (based on metagenomics) along the salinity gradient. We calculated the richness, Shannon index, and Faith's PD of observed OTUs to characterize the taxonomic and phylogenetic alpha diversity, and the richness and Shannon index of KOs to represent the alpha diversity of microbial functional potential. Ordinary least squares linear was performed to test correlations of taxa/KO abundance and alpha diversity with salinity. Segmented linear regression was performed to test the potential breakpoint for archaea taxa abundance across the salinity gradient. The similarity between microbial communities along the salinity gradient was displayed by nonmetric multidimensional scaling (NMDS) using Bray-Curtis distances by "vegan" package[66] in R 3.6.1.

To unravel genomic characteristics of salt-tolerant microorganisms, we respectively compared the genomic information (including genome size, KEGG pathways/KOs abundance per taxon) of taxa in neg-bac vs. pos-bac groups, as well as neg-arch vs. pos-arch groups. Significance was assessed with unpaired two-sided Mann–Whitney U-test using SPSS 20 statistical software (SPSS Inc., Chicago, IL, USA). To investigate differences in KEGG pathways between salt-sensitive and salt-tolerant taxa, KOs were grouped into BRITE hierarchy level 3 pathways. Pathways with a significant absolute abundance difference of >0.001 between the two groups were selected for visualization using heatmaps combined with hierarchical clustering. Additionally, differences in C-acquisition KOs and CAZymes among four response groups were visualized by NMDS based on Bray-Curtis distances. PERMANOVA was performed to test the significant dissimilarity.

Ordinary least squares (OLS) linear regression was used to decipher the variation in abundance of overall KOs per taxon across the salinity gradient. Using partial correlation with EC as the control variable, test the relationship between edaphic factors and microbial genomic traits. To unravel predictor KEGG pathways of salinity responses among microbiomes, the pathway abundance at level 3 was regressed against salinity within each response group using the

function package "randomForest"[67] of *R* (ntree = 1,000). Lists of pathways ranked by RF in order of feature importance were determined over 100 iterations, and 10-fold cross-validation was used to identify the number of biomarker pathways. A phylogenetic tree of taxa in four response groups was visualized with the webtool iTOL (https://itol.embl.de/)[68]. All other graphs were completed with R packages "ggplot2"[69], "circlize"[70] and "UpSetR"[71].

## Reporting summary

Further information on research design is available in the Nature Portfolio Reporting Summary linked to this article.

## Data availability

The raw microbial metagenomic sequencing data derived from this study have been deposited in the NCBI SRA database under BioProject number PRJNA1018220. The raw16S rRNA gene amplicon sequences derived from this study have been deposited in the NCBI SRA database under BioProject number PRJNA1123169. All the materials, raw data, and protocols used in the article are available upon request. The raw data associated with this study have been publicly archived in Figshare at https://doi.org/10.6084/m9.figshare.24065409. Source data are provided with this paper.

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

## Acknowledgements

We are grateful to Prof. Mingxiu Gao and Prof. Yuguo Zhao for their support of soil sampling. R.C. acknowledges the financial support from National Key R&D Program (2022YFD1500304-02). X.Y. acknowledges the financial support from the Agricultural Science and Technology Innovation Program (ASTIP No. CAAS-ZDRW202407) and Youth Innovation of Chinese Academy of Agricultural Sciences (Y2023QC35).

## Author contributions

R.C., Y.F. and E.G. coordinated the overall idea; Y.D., B.Y. and Y.B. conducted the experimental analysis; Y.D. completed statistical analyses and wrote the first draft of the manuscript; R.C., E.G., X.L. and X.Y. provided suggestions; R.C. and E.G. revised the manuscript. All authors contributed critically to the manuscript and gave final approval for publication.

## Competing interests

The authors declare no competing interests.

## Additional information

**Peer review information** : *Nature Communications* thanks the anonymous reviewer(s) for their contribution to the peer review of this work. A peer review file is available.

