## [Peer Review File · Nature Communications]

Eco-evolutionary strategies for relieving carbon limitation under salt stress differ across microbial cladesREVIEWER COMMENTS

Reviewer #1 (Remarks to the Author):

In this manuscript, Dong et al. examine the relationship between the life strategies of archaea and bacteria across a salinity gradient. Using a combination of metagenomics and amplicon sequencing, the authors identify distinct groups of both bacteria and archaea that are positively and negatively associated with increasing salinity. They find distinct patterns with salt-resistant taxa by domain, where bacteria tended to have smaller genomes with increasing salinity, whereas archaea tended to have larger genomes. This relationship suggests that bacteria may have smaller genomes in order to conserve energy under salt stress. Archaea had a greater number of C-acquisition genes under salt-stress, suggesting increased metabolic diversity under stress. These results point towards distinct eco-evolutionary strategies of bacteria and archaea along a salinity gradient and adds greater insight to our understanding of the evolutionary forces of soil bacteria. Overall, I found the paper easy to follow and generally well written. However, I believe there are some issues with this paper both in the methods and framing which make it not acceptable for publication at this time.

A major issue is that many of the conclusions hinge on the metric of genes per taxa, yet the estimation of this metric is based on only taxonomy of scaffolds. By not separating these into bins and checking for completeness and contamination, there is the risk that many of the taxa could be incomplete or highly contaminated. I think if the authors want to use a metric of genes/taxa, they need to generate bins for each taxa and assess the quality of these bins.

I recognize that the AACG metric might address the previous point, but it's not clear when this is being used, as it is not included in the figures. I also found the theory behind the AACG metric hard to follow and could be explained more.

Why were only 200 bacterial and 50 archaeal taxa chosen for each response?

L208-210: Why would the opposite trend might be true in bacteria? This seems like a pretty

major result but does not feel fully explained.

Although I agree with the authors overall point of smaller bacterial genomes being a response to salt-stress, I don't know if what is being observed could be considered "gene loss", since you are looking at different taxa. I think this would be more appropriate if you were using a pangenomic approach where you found fewer genes across the salinity gradient within a genome. Even then, there would need to be some type of temporal component to suggest gene loss—as you might also be observing genome expansion moving in the opposite direction on your gradient.

L134-137: This is just Proteobacteria, but what about all of the other phylum? Do you observe the same effect?

Reviewer #2 (Remarks to the Author):

- What are the noteworthy results?

That different prokaryote lineages adopt divergent approaches with regard to genomic streamlining or functional capacity in response to salt stress.

- Will the work be of significance to the field and related fields? How does it compare to the established literature? If the work is not original, please provide relevant references.

In the soil microbiology literature, several recent sources have explored genomic traits and eco-evolutionary trajectories across across different environmental categories, in some cases with more comprehensive analyses. See:

<https://doi.org/10.1093/femsmc/xtab020>

<https://doi.org/10.1093/femsmc/xtab020>

<https://doi.org/10.1038/s41564-023-01465-0>

<https://doi.org/10.1128/mbio.03584-22>

But these studies do not distinguish the broad response types along bacterial and archaeal lines, to which the current paper may be considered novel. However, I am not familiar with the extent that these have been explored in the marine environment.

- Does the work support the conclusions and claims, or is additional evidence needed?

The work shown is in service to, and supports generally, the conclusions and claims made by the authors. See below for further comments.

- Are there any flaws in the data analysis, interpretation and conclusions? - Do these prohibit publication or require revision?

I am concerned with the authors' initial categorization of prokaryote groups as either positively or negatively associated with the salt stress gradient -- and all subsequent interpretations. As far as I can determine from the methods, the amplicon or shotgun sequence abundances of their organisms are not quantifiable in an absolute sense. For their use of positive or negative linear relationships to screen organisms for salt response, their data cannot determine if the increase or decrease of one organism is due to salt, or to the relative increase or decrease of another. In other words, some lineages may truthfully be equally abundant across the gradient but appear to have a strong preference for one habitat or another based solely on the response of other groups. This means that genomic traits cannot be interpreted towards any particular eco-evolutionary strategy. The authors may have a credible defense of their methods that allows them to identify these relationships, but these must be spelled out concretely in their manuscript. Thus, this issue does not preclude publication, but requires attention.

- Is the methodology sound? Does the work meet the expected standards in your field?

In some cases, I have concerns with the methodology. The filtering of prokaryotes into positive or negative salt-response groups likely produces different numbers of organisms in each group. Some of the figures and results presented show gene or KO abundance, but it is not clear how the values are standardized (whether this is accounted for), and in some cases are not standardized. The total numbers of organisms in each response group are not presented in the main text and it is hard to evaluate results as a consequence.

From recent papers that I have read, a greater intentionality towards gene organization beyond high-level KEGG groups may provide a more insightful interpretation of the genomic features that are important to a system. From this standpoint, I'm not sure the current methodology always meets this standard even if the story is compelling.

- Is there enough detail provided in the methods for the work to be reproduced?

Greater details is needed in some places including on the sample site and collection.

Specific comments:

L71: Which kind of microorganisms should the Red Queen Hypothesis also apply to?

L83-84: How was the salinity gradient established? Please briefly summarize here.

L94: I think prokaryote would be more appropriate here since the authors examined bacteria and archaea but not microbial eukaryotes.

L97-105: How were these clades determined? Could the authors briefly summarize how "different responses" were determined and what this signifies? Negative and positive responses? Were these organisms manually screened or was a common method employed?

L100: Please use the full terms of response groups before abbreviating.

L202-203: What are the main groups of KO's considered to be salt-tolerant? Are sigma factors considered for stress-response signaling?

L202-209: This paragraph as written is not clear. It seems from figure 4 that a greater number of salt-intolerant bacteria have a greater number of all three salt-tolerance genes. Why is this? It's difficult to compare the diversity and specific mechanisms of Figure 4 as currently constructed.

L223-224: The relative abundance metric used here should be clarified. Is this a higher proportion of C-acquisition genes on each genome, or a higher proportion of taxa with C-acquisition genes?

L249: What are the seven C-acquisition mechanisms?

L374-383: A little more detail on the site and its conditions is needed here which I think would be immediately useful to the reader. How has the salinity gradient been established? Is this an experimental manipulation? Salinity as by-product of agriculture? Coastal or tidal? What depth were soils taken from and were they close to any plant root zones? Also relevant is how soil moisture varies with salinity -- was this measured?

L403-413: Please indicate the version of QIIME used and citations. Please also include the appropriate citations for the SILVA database. I am concerned that the authors used an older version of their bioinformatic software (QIIME vs. QIIME2) and reference database (SILVA 132 vs. 138) and recommend updating their procedure with current versions.

L422-441: Please include citations for the these tools.

L490-519: Please cite R package utilized.

L443-454: Without or additional standards, sequence abundances must be assumed to be relative -- therefore the linear relationship of one taxon to salt cannot be disentangled from that of another taxon. The euryarchaeota, for example, may not be positively associated with salt but rather all other groups may be negatively associated. I worry this will affect conclusions between an organism's habitat presence, functional repertoire, and thus it's true eco-evolutionary response to salt.

Figure 2B: What is the y-axis unit?

Figure 4B-E: I think the units should be better standardized here. For D and E, the mechanism presence-absence categories on the x-axis should match. The numeric axis on the black bars (and the order of the items) must also match each other for bacteria and for archaea. This will aid in the comparison of positive and negative-response bacterial groups and -- separately -- for positive and negative-response archaea.

REVIEWER COMMENTS

Reviewer #1 (Remarks to the Author):

In this manuscript, Dong et al. examine the relationship between the life strategies of archaea and bacteria across a salinity gradient. Using a combination of metagenomics and amplicon sequencing, the authors identify distinct groups of both bacteria and archaea that are positively and negatively associated with increasing salinity. They find distinct patterns with salt-resistant taxa by domain, where bacteria tended to have smaller genomes with increasing salinity, whereas archaea tended to have larger genomes. This relationship suggests that bacteria may have smaller genomes in order to conserve energy under salt stress. Archaea had a greater number of C-acquisition genes under salt-stress, suggesting increased metabolic diversity under stress. These results point towards distinct eco-evolutionary strategies of bacteria and archaea along a salinity gradient and adds greater insight to our understanding of the evolutionary forces of soil bacteria. Overall, I found the paper easy to follow and generally well written. However, I believe there are some issues with this paper both in the methods and framing which make it not acceptable for publication at this time.

Response: Thank you very much for your careful review and positive comments on our manuscript. We agree with your concerns and have updated the manuscript accordingly. We provide a point-by-point response to your comments below.

Q1. A major issue is that many of the conclusions hinge on the metric of genes per taxa, yet the estimation of this metric is based on only taxonomy of scaffolds. By not separating these into bins and checking for completeness and contamination, there is the risk that many of the taxa could be incomplete or highly contaminated. I think if the authors want to use a metric of genes/taxa, they need to generate bins for each taxa and assess the quality of these bins.'

Response: We appreciate your comment and fully agree that many of the current conclusions are based on the taxonomy of scaffolds. As suggested, the conclusions are more convincing if bins are used in the analysis.

Binning metagenomic sequences presents technical challenges, particularly in distinguishing gene sequences from different taxa and clustering similar sequences of potentially the same origin into a binning. Environmental samples with complex species composition pose a significant challenge for binning. Therefore, at the moment, binning of environmental samples remains still a difficult problem faced by many researchers. We note there is recent precedent for high-impact conclusions derived from gene scaffolds. Bao et al. (2021), for example used scaffolds to assess functional gene abundances for specific taxa involved in litter decomposition (Bao et al., 2021).

Despite these challenges, we carried out sequences binning. We have attached a file (binning_result.excel) containing the genome bins along with their corresponding quality factors for your reference. Initially, we obtained a total of 177 bins from the shotgun metagenomic sequences. After applying filters for completeness (>80%) and contamination (<10%), only 7 bins were left that met our criteria. This limited number

of bins proved to be insufficient for our analysis, and consequently, we are unable to employ the metric of bins.

We feel that our major conclusion remains robust because the genome size information for all taxa in the response groups came from NCBI database (<https://www.ncbi.nlm.nih.gov/genome/genome/>). This database mainly relies on the whole genome sequencing of pure strains, which is to some extent similar to information provided by binning. Our major conclusion is “Bacteria exhibited reduced genome sizes associated with a depletion of core metabolic genes in response to salinity, while archaea displayed larger genomes and an enrichment of salt resistance genes and also core metabolic genes”. While KOs were derived from the metagenomic sequencing in the current study, genome size derived from the NCBI databased was strongly correlated to the abundance of KOs in both bacteria and archaea. We believe this provide evidence that our results based on the scaffolds are reliable.

Reference:

[1] Bao Y, Dolfing J, Guo Z, et al. Important ecophysiological roles of non-dominant Actinobacteria in plant residue decomposition, especially in less fertile soils[J]. *Microbiome*, 2021, 9(1). DOI:10.1186/s40168-021-01032-x.

Q2. I recognize that the AACG metric might address the previous point, but it's not clear when this is being used, as it is not included in the figures. I also found the theory behind the AACG metric hard to follow and could be explained more.

Response: Sorry that we did not elaborate clearly on $AACG_i$. $AACG_i$ is not the final presented metric in the manuscript, but rather the members of each response group (i.e. overall KOs in Figure 3A, KEGG pathway in Figure 3B and 3C, KOs related to salt-resistance in Figure 4A, KOs related to carbon-fixation in Figure 5A, and KOs related to carbon-degradation in Figure 5B). Since our determination on gene abundance is based on scaffolds, it leads to the fact that the abundance of functional genes will be affected by taxa abundance. Therefore, we at first normalized the abundance of functional genes by taxa abundance. $AACG_i$ is the normalized abundance of the specific functional genes of taxon i (equation (1)). Taking Figure 3A as an example (please see Table. R1 for the schematic diagram), pos-bac box was composed of the normalized abundance of overall KOs of the 200 taxa in that group. A total of 5520 KOs were identified for these 200 taxa, with n in $\sum_{j=1}^n ACG_{ij}$ being 5520.

$$AACG_i = \frac{\sum_{j=1}^n ACG_{ij}}{AT_i} \quad (1)$$

in which, $AACG_i$ is the normalized abundance of each category of genes in taxon i , (i.e., overall KOs, KEGG pathways, KOs involved in salt-resistance, and KOs/genes involved in C-acquisition). ACG_{ij} represents the abundance of the gene j of taxon i . n is the number of genes that perform the corresponding functions. AT_i is the abundance of taxon i .

Thank you for reminding us to realize that our expression is prone to confuse readers. Thus, we rephrased the relevant expression in the manuscript, and revised the

abbreviation of *AACG* to *NACG*. Please see lines 507-524 for details.

Table R1. Schematic diagram for calculating the AACG

	pos_bac taxa	$\sum_{j=1}^{5520} ACG_{ij}$ (total overall KOs abundance)	AT_i (total taxon abundance)	$AACG_i$
1	Actinobacteria_bacterium	70519.7	133056.1	=70519.7/133056.1
2	Aquificales_bacterium	51273.4	104426.5	=51273.4/104426.5
3	Salinimicrobium_xinjiangense	61176.7	111230.3	=61176.7/111230.3
4	Longibacter_salinarum	31605.0	60430.3	=31605.0/60430.3
....
200	candidate_division_KSB1_bacterium	284788.9	557316.7	=284788.9/557316.7

Q3. Why were only 200 bacterial and 50 archaeal taxa chosen for each response?

Response: Many thanks for this comment, and it is really a critical issue. A total of 34898 taxa were identified in 37 samples; of which 7215 both had sufficient abundance for statistical analysis and were correlated to salinity. Because the effect size of many correlations was low, we chose to focus on taxa that exhibited the strongest responses to salinity as representative of life history strategies implemented for salt tolerance. This reduced statistical noise from genomic complexity common to most environmental datasets. According to these different considerations, we set up three processes to perform the screening of taxa. Please find the workflow for establishing the four response groups in Figure S2.

Figure S2. Workflow for establishing four response groups

There are three major steps in the workflow:

(1) Remove the taxa with a low abundance, as a result 15718 taxa remained. The taxa with a total abundance of less than 100 across all the samples were filtered out.

Rationale: Rare species with low abundance are often filtered out primarily to reduce statistical artifacts and enhance the efficiency of the analysis. Furthermore, rare species in the environment have less well-characterized genomes compared to more abundant species. We removed rare species because we were concerned these factors would lead

to spurious results.

(2) Find the responsive species with significant positive or negative correlation with salinity (electric conductivity, EC).

Rationale: In our previous research (Yao, et al., 2020), we found that >90% of the phyla, 75% of the genera and 92% of the OTUs did not respond to salt amendments in a controlled experiment. Yao et al. successfully filtered species to obtain clear response patterns to different treatments than detectable in the full microbiome. So in the current study, we selected the responsive species before further analysis.

(3) Rank taxa in each response group by total abundance and select the top 200 bacteria and the top 50 archaea.

Rationale: Analyzing all the responsive species would be computationally challenging and potentially mute statistical signals due to the complexity of environmental microbiomes. We further screened organisms to overcome these obstacles. In the process of further screening, we need to ensure that the number and total abundance of screened taxa are representative while minimizing the interference caused by excessive species. Here, we provide two pieces of data to support these assumptions. First, in the majority of samples (22 of 37 samples), the total abundance of the selected taxa was greater than 50% of the total abundance of all taxa (Figure 1A). Secondly, we calculated the ratio of total number/abundance of the selected taxa to that of all responsive taxa (Table. R2). The ratio of taxa number ranged from 3.8% to 70.4, and the ratio of taxa abundance ranged from 45.1% to 99.5%. Taking into consideration representativeness of the data and the analytical load, we chose to use this scheme with the top 200 bacteria and the top 50 archaea. It should be noted that the number of screened archaea is only 50 instead of 200, due to the limited number of responsive archaeal in our dataset. Specifically, 497 archaeal taxa showed a positive response to salinity, and only 71 showed a negative response.

Table R2. The ratio of the number/abundance of the selected taxa to that of all responsive taxa

	Bacteria with positive response	Bacteria with negative response	Archaea with positive response	Archaea with negative response
Number of taxa with a significant response	1398	5248	497	71
Number of the selected taxa	200	200	50	50
Ratio (%)	14.3 (200/1398)	3.8 (200/5248)	10.1 (50/497)	70.4 (50/71)
Total abundance of taxa with a significant response	6418625	28057136	1079943	1829390
Total abundance of selected taxa with a significant response	4525610	21189231	1074424	825620

Ratio (%)	70.5 (4525610/6418625)	75.5 (21189231/28057136)	99.5 (1074424/1079943)	45.1 (825620/1829390)
-----------	---------------------------	-----------------------------	---------------------------	--------------------------

References:

[1] Yao T, Chen R, Zhang J, et al. Divergent patterns of microbial community composition shift under two fertilization regimes revealed by responding species [J]. *Applied Soil Ecology*, 2020, 154: 103590.

Q4. L208-210: Why would the opposite trend might be true in bacteria? This seems like a pretty major result but does not feel fully explained.

Response: We agree, and we found that taxa in pos-bac group contained fewer salt-resistance mechanisms compared to neg-bac group (Fig. 4B vs 4C). The most likely reason is that bacteria will greatly shrink redundant genes to reduce energy consumption under the abiotic stressors (Simonsen, 2022). It can be inferred that bacteria tend to streamline even salt-resistant genes under high salinity to reduce energy consumption, i.e. some salt-resistant mechanisms are also redundant.

We also found that the number of taxa containing the mechanism of osmotic solutes synthesis (organic solutes synthesis) greatly reduced in pos-bac group compared to neg-bac group (Fig. 4B vs 4C). Also, there was no significant difference in the abundance of genes involved in the synthesis of osmotic solutes between pos-bac and neg-bac taxa, while for the other two mechanisms K^+ uptake and Na^+ extrusion pos-bac group were significantly higher than the neg-bac group (Fig. 4A). Both indicate that salt-tolerant bacteria prefer to apply K^+ uptake and Na^+ extrusion to resist salinity and give up the pathway of osmotic solutes synthesis. In fact, it has been reported that synthesis of organic solutes is a very energy expensive stress mechanism, which can cause a severe reduction of growth yields (i.e., biomass produced per gram C metabolized) by roughly 90% (Schimel et al., 2007). Thus, it is reasonable for bacteria to streamline this salt-resistance mechanism.

To address this comment, we have added the relevant information to the manuscript, “In this study, salt-tolerant bacteria with small genomes contained a lower diversity of salt-resistance and C-acquisition mechanisms than salt-sensitive bacteria (Fig. 4B, 4C, 5C and 5D). It can be inferred that bacteria tend to streamline redundant salt-resistant genes under high salinity. This is consistent with previous work reporting that environmental stress such as low pH, hyperthermia, drought, and salt caused the loss of functionally redundant genes. In particular, the salt-tolerant bacteria taxa were observed to have a higher tendency to sacrifice their salt-resistance mechanisms in synthesizing organic osmotic solutes (Fig. 4B and 4C). The synthesis of organic osmotic solutes is a very energy expensive stress mechanism, which can cause a reduction of growth yields (i.e., biomass produced per gram C metabolized) by roughly 90% (Schimel et al., 2007)”. Please see lines 291-299.

Additionally, we have rearranged Fig. 4 to make the results clearer. Please see the revised Fig. 4 in the following.

Figure 4. Differences in salt resistance genes among bacterial and archaeal salinity response groups. A. KOs associated with salt tolerance mechanisms per taxon. B to E. Distribution frequencies of taxa containing each salt resistance mechanism in each response group visualized with UpSet plots.

References:

- [1] Simonsen AK. Environmental stress leads to genome streamlining in a widely distributed species of soil bacteria. *Isme Journal* 16, 423-434 (2022).
- [2] Schimel J, Balsler TC, Wallenstein M. Microbial stress-response physiology and its implications for ecosystem function. *Ecology* 88, 1386-1394 (2007).

Q5. Although I agree with the authors overall point of smaller bacterial genomes being a response to salt-stress, I don't know if what is being observed could be considered "gene loss", since you are looking at different taxa. I think this would be more appropriate if you were using a pangenomic approach where you found fewer genes across the salinity gradient within a genome. Even then, there would need to be some type of temporal component to suggest gene loss—as you might also be observing genome expansion moving in the opposite direction on your gradient.

Response: It is really a great comment! We agree with you that smaller genome size is not equal to gene loss. In current study, we aimed to compare the genomic strategies adopted by bacteria and archaea to cope with energy constraints in a high-salt environment. Genome size is an intuitive representation of its strategy, as it can determine the level of energy consumption. Even so, our study is inevitably linked to gene loss and gene expansion. Because when facing the phenomenon that salt-tolerant bacteria possessed with smaller genomes, as well as salt-tolerant archaea possessed with larger genomes, we discussed that they had undergone gene loss/expansion. To provide support, we separately investigated the changes in the abundance of functional genes in bacterial and archaeal group across the salinity gradient. As expected, the total abundance of bacterial functional genes (overall KOs) was significantly reduced along

the salinity gradient, while archaea showed an increase (Figure. R1). We also found that the response of archaeal functional gene abundance to salinity was nonlinear, and there is a salinity threshold. The response rate before the threshold (EC <1.8 dS/m) is greatly higher than that after the threshold (EC > 1.8 dS/m). We have added Figure R1 to the supplementary material and added related information to the manuscript. More details please see lines 194-196, and 277-279.

Figure R1: Changes in the abundance of overall KOs per taxa across the soil salinity gradients of bacterial (A) and archaeal (B) groups

As you mentioned, pan-genomic changes along the salinity gradient are also insufficient to support gene loss/expansion, as their direction of evolution is uncertain. It is informative to observe the gene loss/expansion in the temporal dimension, while referencing space-for-time studies that use spatial gradients is also a common approach when temporal studies are not easy to achieve in an environment. But still, temporal dimension is a very attractive research topic, and we are interested in including it in our subsequent studies. Simonsen et al. (2022) did a similar study, and demonstrated in the temporal dimension that environmental stress leads to a continuous genome reduction along four stress gradients (acidity, aridity, heat, salinity) in naturally occurring populations of *Bradyrhizobium diazoefficiens*. This finding can somehow provide support for bacterial gene loss in our study. Please see lines 293-295.

References:

[1] Simonsen AK. Environmental stress leads to genome streamlining in a widely distributed species of soil bacteria. *Isme Journal* 16, 423-434 (2022).

Q6. L134-137: This is just Proteobacteria, but what about all of the other phylum? Do you observe the same effect?

Response: Sorry that we did not elaborate this point clearly in the previous version of the manuscript. To rule out phylogenetic differences, we compared the genome sizes of taxa belonging to the same phylum between the positive and negative response groups. For bacteria, only Proteobacteria and Chloroflexi were shared in the neg-bac and pos-bac group. The relative abundance of Chloroflexi taxa in the pos-bac and neg-bac groups is very low (<2%, Fig. S6). Moreover, the taxa number of Chloroflexi (10 taxa vs 6 taxa) is too small to support the reliability of the statistical analysis (Fig. 2A). Thus, only Proteobacteria (122 taxa vs 72 taxa) were further analyzed. To avoid causing

confusion, we have supplemented this information to the manuscript. Please see lines 131-135.

Reviewer #2 (Remarks to the Author):

Q1. What are the noteworthy results?

That different prokaryote lineages adopt divergent approaches with regard to genomic streamlining or functional capacity in response to salt stress.

Response: Thank you very much for your recognition and constructive suggestions. We have carefully revised our manuscript according to your suggestion. We believe that our manuscript is greatly improved after the revision.

- Will the work be of significance to the field and related fields? How does it compare to the established literature? If the work is not original, please provide relevant references.

In the soil microbiology literature, several recent sources have explored genomic traits and eco-evolutionary trajectories across across different environmental categories, in some cases with more comprehensive analyses. See:

<https://doi.org/10.1093/femsmc/xtab020>

<https://doi.org/10.1093/femsmc/xtab020>

<https://doi.org/10.1038/s41564-023-01465-0>

<https://doi.org/10.1128/mbio.03584-22>

But these studies do not distinguish the broad response types along bacterial and archaeal lines, to which the current paper may be considered novel. However, I am not familiar with the extent that these have been explored in the marine environment.

Response: Many thanks for your careful review and the relevant references. We have carefully studied the provided articles and their associated ones, which inspired us a lot. The article “Variation in genomic traits of microbial communities among ecosystems” (Chuckran et al., 2021, <https://doi.org/10.1093/femsmc/xtab020>) highlights the macro significance of our research. It “assesses microbial communities through **a trait-based framework** highlights important **relationships between microbes and their environment** which may not be detectable through taxonomic analyses alone” (Chuckran et al., 2021). It underscores the knowledge gap that we address here: “much of this knowledge concerning bacterial genomic traits has been derived from cultures or isolates. This presents substantial bias in our understanding of these relationships, especially for genomic traits in complex microbial communities. An alternative approach is to **examine genomic traits on a community level** in situ. This is an important practice for microbial ecology as there has been growing interest in trait dimensions which might improve our assessment of community function, **yet little work has been done to observe these traits on the community level**” (Chuckran et al., 2021). Our research discovered opposite eco-evolutionary strategies in response to salt stress, based on examining genomic traits on a community level. We added relevant information in the Introduction part. Please see lines 69-71.

This article also points out the technical feasibility of our approach in the current study, highlighting several useful traits independent of binning approaches suggested by

reviewer 1. “Genomic traits such as GC content, number of regulatory genes and average genome size may be especially useful for this purpose, as they can often be easily estimated from metagenomic datasets and do not require an extensive knowledge of the taxa within the community. The relative ease with which these traits may be derived makes them ideal metrics for large-scale comparisons and represents a potentially valuable tool for linking microbial communities with ecosystem-level processes” (Chuckran et al., 2021). In our study, we focused on genome size, abundance of KOs, KEGG pathways, specific KOs/ genes encoding for salt resistance and C-acquisition as genomic traits. We added relevant information in the Introduction. Please see lines 71-73.

Although several recent sources have explored genomic traits and eco-evolutionary strategies across different environmental categories, as noted by the Reviewer, there exists a huge difference between our study and previous studies. Many of the previous studies draw conclusions by analyzing and comparing different genomic (and also taxonomic) traits of one certain microorganism and/or focus on bacteria.

For example in marine communities, the genome size and GC content of bacteria declined in parallel, consistent with genomic streamlining. In contrast, soil communities averaging smaller genomes featured higher GC content. The contrasting trends of genome size and GC content suggests unique selection pressures in soil bacteria versus marine communities (Chuckran et al., 2021).

In other work, Piton et al., 2023 (<https://doi.org/10.1038/s41564-023-01465-0>) used shotgun metagenomes to characterize overarching covariations of the genomic traits that capture dominant life history strategies in bacterial communities. Barnett et al., 2023 (<https://doi.org/10.1128/mbio.03584-22>) identified several genomic features associated with patterns of bacterial C acquisition and growth; notably genomic investment in resource acquisition and regulatory flexibility; genomic trade-offs defined by numbers of transcription factors, membrane transporters, and secreted products, which match predictions from life history theory. Lastly, Wang et al., 2023 found that bacterial average genome size and gene functional diversity decrease, whereas taxonomic diversity increases, as soil pH rises from acidic to neutral. Thus, a mismatch between taxonomic and functional diversity can arise when environmental factors (such as pH) select for adaptive strategies that affect genome size distributions. Unlike mining the consistency or mismatch of genomic traits or genomic/taxonomic traits embedded in the genomic features of bacteria, our current study distinguished the broad response types along bacterial and archaeal lines, which is also pointed out as a novel finding by the Reviewer. Direct comparison of genomic traits between bacteria and archaea is helpful to show divergent patterns in eco-evolutionary adaptations to soil saline stress.

Previous studies have reported that a single source of pressure, such as low pH (Cortez et al., 2022; Ullrich et al., 2016) or high temperature (Sabath et al., 2013; Sorensen et al., 2019), can drive the streamline of bacterial genomes. Our major conclusion, bacteria exhibited reduced genome sizes associated with a depletion of core metabolic genes in

response to salinity, while archaea displayed larger genomes and extended metabolic capacity. We further concluded that bacteria conserve energy through genome streamlining when facing salt stress, while archaea invest in carbon acquisition pathways to broaden their resource usage. Moreover, an emerging report (Wang et al., published in November 2023) indicates that soil bacteria tend to have larger genome sizes and higher functional diversity when soil pH decreases from neutral to acidic. Genomic streamlining can compromise the ability of microorganisms to cope with environmental fluctuations (Barberan et al., 2014), highlighting the need for further research in this area.

Current research in marine microbiology is primarily focused on studying individual species, with limited investigation at the community level. Fewer works existing in soil microbiology A common observation across many of marine studies is the prevalence of streamlined genomes in the researched species, predominantly bacteria (Giovannoni et al., 2005; Schneiker et al., 2006; Giovannoni et al., 2014; Kashtan et al., 2014; Luo et al., 2014; Graham et al., 2021; Roda-Garcia et al., 2021). While some marine species have been found to possess larger genomes (Giovannoni et al., 2014), it appears that species with smaller genomes tend to be more well-researched. This bias stems from the belief that compact genomes are more likely to harbor unique mechanisms for environmental adaptation. Consequently, there exists a gap in our understanding of energy adaptation strategies employed by salt-tolerant microorganisms with larger genomes in marine research.

References

- [1] Piton G, et al. Life history strategies of soil bacterial communities across global terrestrial biomes. *Nature microbiology* 8, 2093-+ (2023).
- [2] Cortez D, Neira G, Gonzalez C, Vergara E, Holmes DS. A Large-Scale Genome-Based Survey of Acidophilic Bacteria Suggests That Genome Streamlining Is an Adaption for Life at Low pH. *Frontiers in Microbiology* 13, (2022).
- [3] Sabath N, Ferrada E, Barve A, Wagner A. Growth Temperature and Genome Size in Bacteria Are Negatively Correlated, Suggesting Genomic Streamlining During Thermal Adaptation. *Genome Biology and Evolution* 5, 966-977 (2013).
- [4] Sorensen JW, Dunivin TK, Tobin TC, Shade A. Ecological selection for small microbial genomes along a temperate-to-thermal soil gradient. *Nature Microbiology* 4, 55-+ (2019).
- [5] Ullrich SR, et al. Gene Loss and Horizontal Gene Transfer Contributed to the Genome Evolution of the Extreme Acidophile "Ferroplasma". *Frontiers in Microbiology* 7, (2016).
- [6] Barberan A, Ramirez KS, Leff JW, Bradford MA, Wall DH, Fierer N. Why are some microbes more ubiquitous than others? Predicting the habitat breadth of soil bacteria. *Ecology Letters* 17, 794-802 (2014).
- [7] Barnett SE, Egan R, Foster B, Eloe-Fadrosh EA, Buckley DH. Genomic Features Predict Bacterial Life History Strategies in Soil, as Identified by Metagenomic Stable Isotope Probing. *mBio*, (2023).
- [8] Chuckran PF, Hungate BA, Schwartz E, Dijkstra P. Variation in genomic traits of microbial communities among ecosystems. *FEMS microbes* 2, xtab020-xtab020 (2021).

- [9] Giovannoni, S. J., Tripp, H. J., Givan, S., Podar, M., Vergin, K. L., Baptista, D., et al. (2005). Genome streamlining in a cosmopolitan oceanic bacterium. *Science* 309, 1242–1245. doi: 10.1126/science.1114057
- [10] Giovannoni, S. J., Cameron Thrash, J., and Temperton, B. (2014). Implications of streamlining theory for microbial ecology. *ISME J.* 8, 1553–1565. doi: 10.1038/ismej.2014.60
- [11] Kashtan, N., Roggensack, S. E., Rodrigue, S., Thompson, J. W., Biller, S. J., Coe, A., et al. (2014). Single-cell genomics reveals hundreds of coexisting subpopulations in wild *Prochlorococcus*. *Science* 344, 416–420. doi: 10.1126/science.1248575
- [12] Schneiker, S., dos Santos, V. A. M., Bartels, D., Bekel, T., Brecht, M., Buhrmester, J., et al. (2006). Genome sequence of the ubiquitous hydrocarbon-degrading
- [13] Graham, E. D., and Tully, B. J. (2021). Marine *Dadabacteria* exhibit genome streamlining and phototrophy-driven niche partitioning. *ISME J.* 15, 1248–1256. doi: 10.1038/s41396-020-00834-5
- [14] Luo, H., Swan, B. K., Stepanauskas, R., Hughes, A. L., and Moran, M. A. (2014). Evolutionary analysis of a streamlined lineage of surface ocean *Roseobacters*. *ISME J.* 8, 1428–1439. doi: 10.1038/ismej.2013.248
- [15] Wang C, et al. Bacterial genome size and gene functional diversity negatively correlate with taxonomic diversity along a pH gradient. *Nature Communications* (2023) 14:7437
- [16] Roda-Garcia JJ, Haro-Moreno JM, Huschet LA, Rodriguez-Valera F, Lopez-Perez M. Phylogenomics of SAR116 Clade Reveals Two Subclades with Different Evolutionary Trajectories and an Important Role in the Ocean Sulfur Cycle. *mSystems* 6, (2021).

Q2. Does the work support the conclusions and claims, or is additional evidence needed?

The work shown is in service to, and supports generally, the conclusions and claims made by the authors. See below for further comments.

Response: Thank you very much for your recognition. We have responded to your comments point-by-point below.

- Are there any flaws in the data analysis, interpretation and conclusions? - Do these prohibit publication or require revision?: I am concerned with the authors' initial categorization of prokaryote groups as either positively or negatively associated with the salt stress gradient -- and all subsequent interpretations. As far as I can determine from the methods, the amplicon or shotgun sequence abundances of their organisms are not quantifiable in an absolute sense. For their use of positive or negative linear relationships to screen organisms for salt response, their data cannot determine if the increase or decrease of one organism is due to salt, or to the relative increase or decrease of another. In other words, some lineages may truthfully be equally abundant across the gradient but appear to have a strong preference for one habitat or another based solely on the response of other groups. This means that genomic traits cannot be interpreted towards any particular eco-evolutionary strategy. The authors may have a credible defense of their methods that allows them to identify these relationships, but these must

be spelled out concretely in their manuscript. Thus, this issue does not preclude publication, but requires attention.

Response: Thank you for this comment. It is really a critical issue. We fully understand your concerns, as the shotgun sequence abundances of microorganisms are not quantifiable in an absolute sense. Therefore, it is difficult to distinguish whether the change in relative abundance of a taxon is caused by changes in its absolute abundance or by offset from other taxa.

While changes in absolute abundance are the gold standard, quantifying the absolute abundance of thousands of microbial species in situ in highly diverse environmental communities is currently impossible. Inferences from studies based on the relative abundance of sequence-based data have driven the field of microbial ecology with valuable insights over the past few decades. Although it is not possible to determine which species are increasing and which are decreasing, we argue that an increase in relative abundance still reflects the influence of a positive ecoevolutionary force, and vice versa for decreases in relative abundance. In this scenario, the organism either persists while others do not (in this study, putatively reflecting a selective advantage conferred by genomic traits) or increases in absolute abundance, where the same inference can be drawn.

At the same time, we fully agree with you that the wrong grouping will interfere with our conclusions. In order to check the effectiveness of the grouping, we did evaluation via two ways.

On one hand, the published salt-tolerant taxa are used as reference standards to observe the groups in which they were included. The published genera of salt-tolerant bacteria mainly include Halo-, Salini-, Thio-, Ocean-, Marin-, Aquisalimonas, Aliifodinibius, Roseovarius, Rubrivirga, Vibrio (Kaitouni et al., 2020; Wang et al., 2013; Montes et al., 2008). These genera do not cover the neg-bac group at all, while they covered 81 taxa in the pos-bac group accounting for 61.8% of the identified genus-level taxa (69 taxa were not identified at the genus level). In addition, the remaining taxa in pos-bac group not included in the salt-tolerant genus are almost all extremophiles existing in marine and alkaline lakes. The published class of salt-tolerant archaea mainly include Halobacteria and Halophilic methanogenic archaea (Oren et al., 2014). They were never included in the neg-arch group at all, but 100% covered the pos-arch group. This is also reflected on the result that the genome size of pos-arch taxa (50 taxa) overlaps with that of published salt-tolerant archaeal taxa (1603 taxa) (Figure 2D).

On the other hand, the abundance of salt-resistance genes is used as an important index to test the grouping effectiveness. For both bacteria and archaea, the abundance of salt-tolerant genes in the salt-sensitive group was significantly lower than that in the salt-tolerant group (Figure 4A). Therefore, it is reasonable to believe that the grouping is valid and does not interfere with our conclusion. We have added the critical information to the manuscript. Please see lines 204-205 and lines 469-479 for details.

Reference:

[1] Kaitouni LBD, Anissi J, Sendide K, El Hassouni M. Diversity of hydrolase-producing halophilic bacteria and evaluation of their enzymatic activities in submerged

cultures. *Annals of Microbiology* 70, (2020).

[2] Wang Y-X, et al. *Aliifodinibius roseus* gen. nov., sp nov., and *Aliifodinibius sediminis* sp nov., two moderately halophilic bacteria isolated from salt mine samples. *International Journal of Systematic and Evolutionary Microbiology* 63, 2907-2913 (2013).

[3] Montes MJ, Bozal N, Mercade E. *Marinobacter guineae* sp nov., a novel moderately halophilic bacterium from an Antarctic environment. *International Journal of Systematic and Evolutionary Microbiology* 58, 1346-1349 (2008).

[4] Oren A. Taxonomy of halophilic Archaea: current status and future challenges. *Extremophiles* 18, 825-834 (2014).

Q3- Is the methodology sound? Does the work meet the expected standards in your field?

In some cases, I have concerns with the methodology. The filtering of prokaryotes into positive or negative salt-response groups likely produces different numbers of organisms in each group. Some of the figures and results presented show gene or KO abundance, but it is not clear how the values are standardized (whether this is accounted for), and in some cases are not standardized. The total numbers of organisms in each response group are not presented in the main text and it is hard to evaluate results as a consequence.

Response: Sorry for our vagueness in the manuscript. In this study, we have normalized the abundance of functional genes by taxa abundance. This ensures that the abundance of the functional genes we studied is not interfered by the abundance of taxa. In particular, our normalization is specific to each taxon. It is achieved by the ratio of the abundance of specific functional genes to the abundance of corresponding taxon ($NACG_i$, (equation (1)). Taking Figure 3A as an example (see Fig. R1 for the schematic diagram), pos-bac box was composed of the normalized abundance of overall KOs of the 200 taxa in that group. A total of 5520 KOs were identified for these 200 taxa, with n in $\sum_{j=1}^n ACG_{ij}$ being 5520. We have rephrased the relevant expression in the manuscript. Please see lines 507-524 for details.

$$NACG_i = \frac{\sum_{j=1}^n ACG_{ij}}{AT_i} \quad (1)$$

in which, $NACG_i$ is the normalized abundance of each category of genes in taxon i , (i.e., overall KOs, KEGG pathways, KOs involved in salt-resistance, and KOs/genes involved in C-acquisition). ACG_{ij} represents the abundance of the gene j of taxon i . n is the number of genes that perform the corresponding functions. AT_i is the abundance of taxon i .

Table R1. Schematic diagram for calculating the NACG

	pos_bac taxa	$\sum_{j=1}^{5520} ACG_{ij}$ (total overall KOs abundance)	AT_i (total taxon abundance)	$NACG_i$
1	Actinobacteria_bacterium	70519.7	133056.1	=70519.7/133056.1
2	Aquificales_bacterium	51273.4	104426.5	=51273.4/104426.5
3	Salinimicrobium_xinjiangense	61176.7	111230.3	=61176.7/111230.3
4	Longibacter_salinarum	31605.0	60430.3	=31605.0/60430.3
....
200	candidate_division_KSB1_bacterium	284788.9	557316.7	=284788.9/557316.7

-Q4: From recent papers that I have read, a greater intentionality towards gene organization beyond high-level KEGG groups may provide a more insightful interpretation of the genomic features that are important to a system. From this standpoint, I'm not sure the current methodology always meets this standard even if the story is compelling.

Response: We fully agree with your opinion. The KEGG is widely used in the field of metagenomics for functional annotation and analysis of microbial communities. It provides a well-curated collection of biological pathways and a common framework for interpreting metagenomic data. Thus, the KEGG pathway is one aspect we are focusing on in the current study. In addition to KEGG pathway, the CAZymes database (Figure 5) were applied, which has specific annotation of functional potential of microbial communities with respect to carbohydrate utilization and degradation. It is useful to characterize microbial strategies for relieving carbon limitation under salt stress, which is one of our major concerns. Our study also investigated specific functional traits based on KOs related to resistance to salt stress (Figure 4) and CO₂ fixation (Figure 5).

- Q5: Is there enough detail provided in the methods for the work to be reproduced?

Greater details is needed in some places including on the sample site and collection.

Response: Many thanks for the suggestion. We have added the detailed information of sample collection in the supplementary (table S4), including longitude (*E*), latitude (*N*), land type, vegetation and soil properties.

Specific comments:

L71: Which kind of microorganisms should the Red Queen Hypothesis also apply to?

Response: Sorry for our omission. We have revised the relevant sentence as “However, insights from marine microorganisms suggest that the RQH may apply to some prokaryote microorganisms”

L83-84: How was the salinity gradient established? Please briefly summarize here.

Response: Sorry for the simplification. The soil salinity gradient is established based on the soil electric conductivity (EC). We added the important information in the manuscript. Please see lines 76-77.

L94: I think prokaryote would be more appropriate here since the authors examined bacteria and archaea but not microbial eukaryotes.

Response: Done! Please see line 86.

L97-105: How were these clades determined? Could the authors briefly summarize how "different responses" were determined and what this signifies? Negative and positive responses? Were these organisms manually screened or was a common method employed?

Response: Thank you for this comment. We provide details on the grouping method and screening of the four groups neg-bac, pos-bac and pos-arch in the response to Q3 of the first Reviewer. We also expanded this portion of the methods to provide more detail. We rephrased it as "To investigate possible eco-evolutionary strategies of salt-tolerant bacteria and archaea, we screened 500 taxa with positive and negative responses to salinity (see Method section for details), which are dispersed in bacterial and archaeal clades. These 500 taxa formed four responsive groups, specifically, bacteria with negative response to salinity (neg-bac), bacteria with positive response to salinity (pos-bac), archaea with negative response to salinity (neg-arch), and archaea with positive response to salinity (pos-arch), respectively (Figure S2)." Please see lines 92-99 of manuscript for more details.

L100: Please use the full terms of response groups before abbreviating.

Response: Done! "Specifically, there were bacteria with negative response to salinity (neg-bac), bacteria with positive response to salinity (pos-bac), archaea with negative response to salinity (neg-arch), and archaea with positive response to salinity (pos-arch), respectively (Fig. S2)". Please see lines 95-98.

L202-203: What are the main groups of KO's considered to be salt-tolerant? Are sigma factors considered for stress-response signaling?

Response: Sorry that we did not include the sigma factors in this study since they have not been reported in archaea so far. In this study, we mainly focused on the three dominant adaptive mechanisms for microorganisms to cope with salinity: (1) pumping Na^+ out of the cell through Na^+/H^+ antiporters to maintain the homeostasis of intracellular Na^+ (Ventosa et al., 1998), (2) accumulating K^+ via the K^+ transport system or K^+/H^+ antiporter (Ventosa et al., 1998), and (3) absorbing and/or synthesizing low molecular compounds to resist osmotic pressure (Kempf et al., 1998). Microbial KOs related to each of these mechanisms are listed in Table S6. We have added this information searchable in lines 208-209, and lines 492-496.

Here, we also complemented the functional gene abundance of sigma factors for the two bacterial response groups. The result showed that taxa of neg-bac with the larger genome size had a higher abundance of functional genes for sigma factor (Figure R2, $P < 0.001$). Sigma factor is a non-specific stress factor, which assists microorganisms in adapting to environmental fluctuations, such as high temperature stress, high osmotic stress, heavy metal stress. However, when it comes to specific salt stress, streamlined salt-tolerant bacteria tend to employ specific salt-tolerance mechanisms to increase

energy use efficiency and reduce energy consumption. For more details, please refer to the response to the following specific comment L202-209.

Figure R2. Differences in sigma-factor genes between the two bacterial groups

Reference:

- [1] Ventosa A, Nieto JJ, Oren A. Biology of moderately halophilic aerobic bacteria. *Microbiology and Molecular Biology Reviews* 62, 504-+ (1998).
- [2] Kempf B, Bremer E. Uptake and synthesis of compatible solutes as microbial stress responses to high-osmolality environments. *Archives of Microbiology* 170, 319-330 (1998).

L202-209: This paragraph as written is not clear. It seems from figure 4 that a greater number of salt-intolerant bacteria have a greater number of all three salt-tolerance genes. Why is this? It's difficult to compare the diversity and specific mechanisms of Figure 4 as currently constructed.

Response: Many thanks for this suggestion. We found that taxa in pos-bac group contain fewer salt-resistance mechanisms compared to neg-bac group (Fig. 4B vs 4C), and this description was added in this paragraph (Lines 211-212). These results are consistent with a scenario in which bacteria sacrifice functional redundancy for a more limited set of salt-tolerance mechanisms when confronted with salt-stress. They support genomic streamlining of salt-resistant genes under high salinity to reduce energy consumption, congruent with work by Simonsen (2022) demonstrating that bacteria can greatly shrink redundant genes to reduce energy consumption under the abiotic stressors

We also found support for the reduction of osmolyte production in particular as a pathway for genome streamlining. The number of taxa containing the mechanism of osmotic solutes synthesis (organic solutes synthesis) greatly reduced in pos-bac group compared to neg-bac group (Fig. 4B vs 4C). Also, there was no significant difference in the abundance of genes involved in the synthesis of osmotic solutes between pos-bac and neg-bac taxa, while for the other two mechanisms K^+ uptake and Na^+ extrusion pos-bac group were significantly higher than the neg-bac group (Fig. 4A). Both indicate that salt-tolerant bacteria prefer to apply K^+ uptake and Na^+ extrusion to resist salinity and give up the pathway of osmotic solutes synthesis. It has been reported that synthesis of organic solutes is a very energy expensive stress mechanism, which can cause a

severe reduction of growth yields (i.e., biomass produced per gram C metabolized) by roughly 90% (Schimel et al., 2007). Thus, it is reasonable for bacteria to reduce genes for osmolyte production as a salt-resistance mechanism.

To more clearly explain this rationale, we have supplemented the relevant information in the manuscript, “In this study, salt-tolerant bacteria with small genomes contained a lower diversity of salt-resistance and C-acquisition mechanisms than salt-sensitive bacteria (Fig. 4B, 4C, 5C and 5D). Consistent with previous work reporting that environmental stress such as low pH, hyperthermia, drought, and salt caused the loss of functionally redundant genes (Sabath et al., 2013; Sorensen et al., 2019; Verma et al., 2022; Ullrich et al., 2016; Zhang et al., 2017), we therefore propose that bacteria tend to streamline redundant salt-resistant genes under high salinity. In particular, the salt-tolerant bacteria taxa also seemed to have a higher tendency to sacrifice the synthesis of osmolytes (Fig. 4B and 4C). Because osmolytes are a very energy expensive stress mechanism that can cause a reduction of growth yields (i.e., biomass produced per gram C metabolized) by roughly 90% (Schimel et al., 2007)”. Please see lines 291-299.

Additionally, we have rearranged Figure 4 to make the results clearer. Please see the revised Figure 4 in the following.

References:

- [1] Simonsen AK. Environmental stress leads to genome streamlining in a widely distributed species of soil bacteria. *ISME Journal* 16, 423-434 (2022).
- [2] Schimel J, Balsler TC, Wallenstein M. Microbial stress-response physiology and its implications for ecosystem function. *Ecology* 88, 1386-1394 (2007).
- [3] Sabath N, Ferrada E, Barve A, Wagner A. Growth Temperature and Genome Size in Bacteria Are Negatively Correlated, Suggesting Genomic Streamlining During Thermal Adaptation. *Genome Biology and Evolution* 5, 966-977 (2013).
- [4] Sorensen JW, Dunivin TK, Tobin TC, Shade A. Ecological selection for small microbial genomes along a temperate-to-thermal soil gradient. *Nature Microbiology* 4, 55-+ (2019).
- [5] Verma D, Kumar V, Satyanarayana T. Genomic attributes of thermophilic and hyperthermophilic bacteria and archaea. *World Journal of Microbiology & Biotechnology* 38, (2022).
- [6] Ullrich SR, et al. Gene Loss and Horizontal Gene Transfer Contributed to the Genome Evolution of the Extreme Acidophile "Ferroplasma". *Frontiers in Microbiology* 7, (2016).
- [7] Zhang X, et al. Adaptive Evolution of Extreme Acidophile *Sulfobacillus* thermosulfidooxidans Potentially Driven by Horizontal Gene Transfer and Gene Loss. *Applied and Environmental Microbiology* 83, (2017).

L223-224: The relative abundance metric used here should be clarified. Is this a higher proportion of C-acquisition genes on each genome, or a higher proportion of taxa with C-acquisition genes?

Response: Sorry for the ambiguity. The “relative abundance” of here is “the proportion of C-acquisition genes on overall KOs”. We have rephrased this sentence as “Salt-

tolerant bacteria and archaea tended to contain a higher proportion of C-acquisition genes (relative gene abundance, the proportion of C-acquisition genes on each genome) than their salt-sensitive counterparts (Fig. 5A).” Please see line 226.

L249: What are the seven C-acquisition mechanisms?

Response: The seven C-acquisition mechanisms are reductive pentose phosphate cycle (Calvin cycle), reductive citrate cycle (Arnon-Buchanan cycle), 3-Hydroxypropionate bi-cycle, hydroxypropionate-hydroxybutyrate cycle, dicarboxylate-hydroxybutyrate cycle, reductive acetyl-CoA pathway (Wood-Ljungdahl pathway), and incomplete reductive citrate cycle. We have made this information searchable in lines 252 and 257, and lines 499-506.

L374-383: A little more detail on the site and its conditions is needed here which I think would be immediately useful to the reader. How has the salinity gradient been established? Is this an experimental manipulation? Salinity as by-product of agriculture? Coastal or tidal? What depth were soils taken from and were they close to any plant root zones? Also relevant is how soil moisture varies with salinity -- was this measured?

Response: Thank you for the suggestion. The samples we collected were non-rhizosphere coastal soils with natural salinity. This requires us to sample along the coastal distance gradient and avoid collecting samples from vegetation areas. The salinity gradient was established according to soil electric conductivity (EC), which is ranging from 0.14 dS m⁻¹ to 13.65 dS m⁻¹ (Fig. S1). Due to EC was determined by dried coastal soils, thus soil moisture did not affect the establish of salinity gradient. We have added the relevant information in manuscript “Based on the distance from the coastline and the type of the covered vegetation, a total of 37 topsoil samples were collected. For each site, 20 cores of topsoil (0-20 cm) were abstracted from a 10*10 (m²) quadrat with a serpentine sampling method, using a 30-mm-diameter gouge auger. Notably, the soils studied were non-rhizosphere soils, so vegetated areas should be avoided when collecting samples. Each sample was homogenized and sieved (<2 mm) prior to splitting into two subsamples. According to the EC of the dried soils (ranging from 0.14 dS m⁻¹ to 13.65 dS m⁻¹), the salinity gradient of these soil samples was formed (Fig. S1). Subsamples used for DNA extraction were stored at -40°C, and the others were air-dried for chemical analysis. The detailed sites description and edaphic properties from the 37 samples were supplied in Table S4, and have previously been reported by Dong et al. (2022)”. Please see lines 383-396.

L403-413: Please indicate the version of QIIME used and citations. Please also include the appropriate citations for the SILVA database. I am concerned that the authors used an older version of their bioinformatic software (QIIME vs. QIIME2) and reference database (SILVA 132 vs. 138) and recommend updating their procedure with current versions.

Response: Thank you for the suggestion. We have re-clustered and re-aligned the sequences with QIIME2 and SILVA 138. It is found that the variation of α - and β -diversity of the newly obtained matrix (Figure R3) in salinity gradient are consistent

with the previous one (Figure R4).

Figure R3. α - and β -diversity based on QIIME 2 and SILVA 138

Figure R4. α - and β -diversity based on QIIME 1.9 and SILVA 132

However, we found many ASVs with considerable abundance that were not able to be assigned to a taxon at the phylum level (Figure R4). This phenomenon did not exist with SILVA 132 (Figure R5). Considering all results associated with species annotation in the main text are based on metagenomic data and that the 16S rRNA amplicon data was used for supporting data only (Figure S3 and Figure S4), we retained the application of QIIME 1.9 and SILVA 132.

Figure R5. Species annotation of ASVs at the phylum level based on SILVA 138

Figure R6. Species annotation of OTUs at the phylum level based on SILVA 132

L422-441: Please include citations for these tools.

Response: Done! Please see lines 429-455.

L490-519: Please cite R package utilized.

Response: Done! Please see lines 536, 552 and 556.

L443-454: Without or additional standards, sequence abundances must be assumed to be relative -- therefore the linear relationship of one taxon to salt cannot be disentangled from that of another taxon. The euryarchaeota, for example, may not be positively associated with salt but rather all other groups may be negatively associated. I worry this will affect conclusions between an organism's habitat presence, functional repertoire, and thus it's true eco-evolutionary response to salt.

Response: Thank you for the comment. Please see our response to Q2.

Figure 2B: What is the y-axis unit?

Response: We apologize for the oversight. The unit of y-axis is Mbp, and we have added it in the graph.

Figure 4B-E: I think the units should be better standardized here. For D and E, the mechanism presence-absence categories on the x-axis should match. The numeric axis on the black bars (and the order of the items) must also match each other for bacteria

and for archaea. This will aid in the comparison of positive and negative-response bacterial groups and -- separately -- for positive and negative-response archaea.

Response: Done! For comparison purposes, we have unified the scales for both bacteria and archaea. At the same time, the order of the three salt tolerance mechanisms is also kept consistent (Figure 4).

Figure 4. Differences in salt-resistance genes among bacterial and archaeal salinity response groups

REVIEWER COMMENTS

Reviewer #1 (Remarks to the Author):

The authors addressed all previous comments. I only a major comment and some minor comments.

There are a lot of claims about carbon-acquisition, but were there any measurements on carbon? I understand this plays into a salinity response, but I think other edaphic properties need to be considered here. There are a lot of soil properties that might also influence these trends, and it would be important to show these data if available. Do any of the changes observed here also correlate with the abundance of C, pH, or other nutrient availability? And if so, how strong are these patterns?

Minor:

L274-276: I would suggest rephrasing this. Currently, it sounds like intergenic regions did not influence genome size here, but that could not be determined with these data. I would just rephrase to say that it could not be determined, but you found a significant relationship with functional gene abundance that points towards functional gene loss.

L377: I would replace “abstracted” with “extracted” or “collected”

L455: Capitalize “we”

Fig 5A. I think the chord diagram here is a little bit confusing and a simple boxplot like in Fig 5B could suffice. I think the problem with the chord diagram is that it makes it difficult to compare the positive and negative groups.

Reviewer #2 (Remarks to the Author):

Previous reviewer comments addressed the issue of making positive vs. negative salt-response groups from relative abundance data. In response, the authors evaluated their grouping in two ways: 1) checking both response groups for known salt-tolerant lineages, and 2) the abundance of salt resistance genes in each group. In both cases, the authors found that these distinctions supported their initial grouping.

Another previous comment raised the issue of how, when, and where gene abundances were normalized across the different salt-response groups which had different numbers of organisms assigned to them. In response, the authors have changed or clarified figure axes to show that they are enumerating gene or KO abundance per taxon. However, I still believe it would be useful for the reader's understanding of their groupings for the authors to present the final number of taxa in each of the four categories. The information is currently only available by compiling the data of the 500 screened taxa from their supplemental Table S5. I think the final numbers should be presented on lines 95-97 (and again in the methods for the sake of completeness). This is especially needed because the screening methods don't make it easy to understand how many organisms are considered. In lines 464-467, the authors rank (by abundance) the "top 200 bacteria and top 50 archaea" in each category but produce 500 taxa total.

A previous comment raised the issue of understanding the sampling site and a further line-comment asked about the establishment of the salinity gradient. The reviewer was concerned with understanding about the frequency, duration, and nature of salt intensity. The authors have provided more information to their system (coastal) in the methods and supplemental information. The authors should clarify in the methods section how their EC values were measured (which instrument?). Relatedly, on line 76-77, the authors included information stating that the salinity gradient was established by the soil electrical conductivity. While I appreciate the authors' clarification here, there seems to be some misunderstanding as to the intent of the request. The request was made so that the reader could understand the (very broad) ecological nature of the salinity gradient (coastal). The authors should rephrase so as to best inform their readers that the salinity gradient was established by proximity to a saltwater body at a coastal field site, as measured by EC. Additionally, the previous comments about the relationship between soil moisture and salinity were not centered on measurement alone, and I apologize that this was not more clear. The question about salinity-moisture covariance was about the system in general, as soils closer to the water line should be saltier and also should have more moisture.

Specific comments:

L207-208: Please list out the three salt-tolerant mechanisms here (they can be summarized as in Figure 4).

L227: For consistency with Figure 5, C-degradation genes should also be identified as CAZymes here.

REVIEWER COMMENTS

Reviewer #1 (Remarks to the Author):

The authors addressed all previous comments. I only a major comment and some minor comments.

Response: Thank you very much for your careful review and your constructive comments. We have revised the manuscript accordingly, and provide a point-by-point response to your comments below.

There are a lot of claims about carbon-acquisition, but were there any measurements on carbon? I understand this plays into a salinity response, but I think other edaphic properties need to be considered here. There are a lot of soil properties that might also influence these trends, and it would be important to show these data if available. Do any of the changes observed here also correlate with the abundance of C, pH, or other nutrient availability? And if so, how strong are these patterns?

Response: Thank you for this comment. We fully agree with your concerns, and the previous version of the manuscript did not report any analysis of correlations between environmental factors and genomic traits. To address this concern, we conducted (1) correlation analysis between soil salinity (electric conductivity, EC) and other key edaphic factors and (2) partial correlation between edaphic factors and bacterial or archaeal genomic traits (i.e., overall KO abundance and C-acquisition gene abundance) when controlling for variance already explained by salinity (i.e., EC as the control variable). Please see results in the figure and table below. Briefly, no significant correlations were observed between SOC and EC, DOC and EC (Fig. R1, $P>0.05$); while total nitrogen (TN)、 total phosphorus (TP)、 available nitrogen (AN) and pH had significantly negative correlation with EC (Fig. R1, $P<0.05$). There was also no significant correlation between edaphic factors and microbial genomic traits (Table S7, $P>0.05$). These proved that edaphic factors such as soil C and pH were not the key factors influencing genomic traits in this study. It is worth noting that nitrogen could be a potential factor in influencing carbon acquisition traits beyond salinity (Table S7), but this is beyond the scope of our work.

Figure R1. Pearson correlation analysis between soil electric conductivity (EC) and other abiotic factors.

Table S7. Partial correlation between edaphic factors and bacterial or archaeal genomic traits EC as the control variable

Dependent variable	Independent variable	Control variable	P-value
Abundance of overall KOs of bacterial taxa (normalized by taxa abundance)	SOC	EC	0.122
	DOC	EC	0.370
	pH	EC	0.171
	TN	EC	0.454
	AN	EC	0.290
	TP	EC	0.461
Abundance of C-acquisition genes of bacterial taxa (normalized by taxa abundance)	SOC	EC	0.873
	DOC	EC	0.963
	pH	EC	0.344
	TN	EC	0.088
	AN	EC	0.057
	TP	EC	0.353
Abundance of overall KOs of archaeal taxa (normalized by taxa abundance)	SOC	EC	0.296
	DOC	EC	0.108
	pH	EC	0.633
	TN	EC	0.741
	AN	EC	0.389
	TP	EC	0.463
Abundance of C-acquisition genes of archaeal taxa (normalized by taxa abundance)	SOC	EC	0.490
	DOC	EC	0.283
	pH	EC	0.551
	TN	EC	0.069
	AN	EC	0.855
	TP	EC	0.131

Additionally, our grouping of microorganisms by salinity appears to be robust (Fig. S2), as the vast majority of taxa in pos-bac and pos-arch groups were obtained only in positive response groups and were highly distinguished from salt-tolerant taxa in negative response groups. This suggests that the variance in salt tolerance plays a pivotal role in differentiating genomic traits among the various response groups.

To address this reviewer's concern, we have added Table S7 in the supplementary materials. We also added relevant content in the Methods "The natural differences in salt tolerance among the response groups provided evidence that salinity was a key driver of genomic trait divergence, and it was further supported by the results of partial correlation between edaphic factors and bacterial or archaeal genomic traits EC as the control variable (Table S7)". Please see lines 474-477.

Minor:

L274-276: I would suggest rephrasing this. Currently, it sounds like intergenic regions did not influence genome size here, but that could not be determined with these data. I would just rephrase to say that it could not be determined, but you found a significant relationship with functional gene abundance that points towards functional gene loss.

Response: We agree. We rephrased this sentence to read "The deletion of intergenic regions has been reported to contribute to the reduction in genome size³³, which could not be ascertained from our data. However, in this study, changes in the overall abundance of KOs were consistent with changes in genome size, strongly suggesting a pattern of protein-encoding gene loss leading to genome reduction. Please see lines 279-282.

L377: I would replace "abstracted" with "extracted" or "collected"

Response: We replaced "abstracted" with "extracted". Please see line 386 of the manuscript.

L455: Capitalize "we"

Response: Done.

Fig 5A. I think the chord diagram here is a little bit confusing and a simple boxplot like in Fig 5B could suffice. I think the problem with the chord diagram is that it makes it difficult to compare the positive and negative groups.

Response: Thank you for the suggestion. We have changed Fig. 5A into a boxplot as follows.

Figure 5. Differences in bacterial and archaeal C-acquisition genes in response to salinity. A. Proportion of C-acquisition genes relative to overall gene abundance. B. Per taxon abundance of genes associated with C-acquisition. C. Frequency distribution of genes related to seven main mechanisms for C-fixation across taxa. D. Frequency distribution of genes related to six types of CAZymes for C-degradation across taxa. Total C-acquisition is the sum of CO₂ fixation gene and CAZyme abundances. Boxplots indicate median (middle line), 25th, 75th percentile (box) and 5th and 95th percentile (whiskers). The box consists of the average abundance of genes of 200 or 50 taxa (N=200 for bacteria and N=50 for archaea). The asterisk above the column denotes significant difference (***) $P < 0.001$ and (**) $P < 0.01$ and ns denotes no significant difference ($P > 0.05$).

Reviewer #2 (Remarks to the Author):

Previous reviewer comments addressed the issue of making positive vs. negative salt-response groups from relative abundance data. In response, the authors evaluated their grouping in two ways: 1) checking both response groups for known salt-tolerant lineages, and 2) the abundance of salt resistance genes in each group. In both cases, the authors found that these distinctions supported their initial grouping.

Another previous comment raised the issue of how, when, and where gene abundances were normalized across the different salt-response groups which had different numbers of organisms assigned to them. In response, the authors have changed or clarified figure axes to show that they are enumerating gene or KO abundance per taxon.

Response: Thank you for your careful review and your constructive suggestions in the following. We have revised the manuscript accordingly, and provide a point-by-point response to your comments below.

However, I still believe it would be useful for the reader's understanding of their groupings for the authors to present the final number of taxa in each of the four

categories. The information is currently only available by compiling the data of the 500 screened taxa from their supplemental Table S5. I think the final numbers should be presented on lines 95-97 (and again in the methods for the sake of completeness). This is especially needed because the screening methods don't make it easy to understand how many organisms are considered. In lines 464-467, the authors rank (by abundance) the "top 200 bacteria and top 50 archaea" in each category but produce 500 taxa total.

Response: Thank you very much for this suggestion. We ranked (by abundance) the top 200 taxa for each group of bacteria and the top 50 taxa were for each group of archaea. That is why a total of 500 taxa was in further investigation. To improve clarity, we have added the requested information in the manuscript (lines 94-96). We state “Specifically, there were bacteria with negative response to salinity (neg-bac, 200 taxa), bacteria with positive response to salinity (pos-bac, 200 taxa), archaea with negative response to salinity (neg-arch, 50 taxa), and archaea with positive response to salinity (pos-arch, 50 taxa), respectively, for a total of 500 taxa”. We also added the following sentence to lines 459-461: “The top 200 bacteria in each bacterial group, and the top 50 archaea in each archaeal group were selected for further investigation, which added up to a total of 500 taxa”.

A previous comment raised the issue of understanding the sampling site and a further line-comment asked about the establishment of the salinity gradient. The reviewer was concerned with understanding about the frequency, duration, and nature of salt intensity. The authors have provided more information to their system (coastal) in the methods and supplemental information. The authors should clarify in the methods section how their EC values were measured (which instrument?).

Response: Thank you for your suggestion. We added more detailed information for EC determination in the manuscript, “EC was determined using electronic conductivity meter (Mettler Toledo, OH, USA) for suspended soils at a soil-water ratio of 1:5 (w/v)”. Please see lines 389-390.

The sampling sites in the current study were in a coastal zone located in the Yellow River Delta, a low gradient floodplain and the transitional zone of the Bohai Sea and the North China plain. In general, the region is easily influenced by salts from seawater because of low elevation. The high ratio of evaporation to precipitation and salt-loaded aerosols are reasons for top soil salinization as well. A study explored the distribution of soil salinity in the Yellow River Delta based on 150 soil samples that were collected in June 2010 (Yu et al., 2014). The results showed that the soil salinity ranged from 0.11 to 10.50 dS m⁻¹ and the salinity in topsoil was higher than that in subsoil. In addition, landforms, land uses, soil types and soil texture were important factors affecting soil salinity. The current distribution of soil salinity in the Yellow River Delta resulted from the comprehensive effects of anthropogenic activities and natural processes. We have added some information and a citation for further details in the M&M, “Soil salinization

in this region resulted from the combined effects of natural processes and anthropogenic activities, such as landform change, climate, and land uses⁴⁹". Please see lines 382-384.

Reference:

Yu, J., Li, Y., Han, G., Zhou, D., Fu, Y., Guan, B., . . . Wang, J. (2014). The spatial distribution characteristics of soil salinity in coastal zone of the Yellow River Delta. *Environmental Earth Sciences*, 72(2), 589-599. doi:10.1007/s12665-013-2980-0

Relatedly, on line 76-77, the authors included information stating that the salinity gradient was established by the soil electrical conductivity. While I appreciate the authors' clarification here, there seems to be some misunderstanding as to the intent of the request. The request was made so that the reader could understand the (very broad) ecological nature of the salinity gradient (coastal). The authors should rephrase so as to best inform their readers that the salinity gradient was established by proximity to a saltwater body at a coastal field site, as measured by EC.

Response: Thank you for the clarification. We apologize for the confusion. We now directly state: "To test these hypotheses, we collected soils along a salinity gradient mainly established by proximity to the coastline and measured by EC (Fig S1), combined shotgun metagenomic and amplicon sequencing...", and "There was a significantly negative correlation between EC and the distance of the sampling site from the coastline ($P < 0.01$, Figure S1)". Please see lines 73-74 and 390-392.

In order to highlight the relationship of EC and the coastal line, we added also Figure R2 and R3 to Figure S1 in the supplementary materials. We also added corresponding information in Introduction and M&M.

Lastly, in order to highlight this ecological significance, we changed the first sentence of the abstract as "With the continuous expansion of saline soils under climate change, understanding the eco-evolution tradeoff between microbial mitigation of carbon limitation and maintenance of functional traits in saline soils, represents a significant knowledge gap in predicting future soil health and ecological function". Please see lines 5-8.

Figure R2. Pearson correlation between EC (representing soil salinity) and the distance

of the sampling site from the coastline. Near Dist function in ArcGIS 10.7 software was used calculate the shortest distance from the sampling site to the coastline.

Figure R3. The spatial distribution of soil salinity near coastal regions. The figure was reproduced from Yu et al. (2014) and encompassed our sampling locations.

Figure S1. Sampling sites for the 37 soil samples and their salinity distribution. A. Map of the area around Shandong Province. B. Distribution map of sampling sites in Dongying. C. Pearson correlation between EC (representing soil salinity) and the distance of the sampling site from the coastline. D. The spatial distribution of soil salinity near coastal regions (reproduced from Yu et al. (2014) and encompassed our sampling locations). Near Dist function in ArcGIS 10.7 software was used calculate the shortest distance from the sampling site to the coastline.

Reference:

Chen, J., & Mueller, V. (2018). Coastal climate change, soil salinity and human migration in Bangladesh. *Nature Climate Change*, 8(11), 981-+. doi:10.1038/s41558-018-0313-8

- Salehin, M. et al. (2018). Mechanisms and Drivers of Soil Salinity in Coastal Bangladesh. In: Nicholls, R., Hutton, C., Adger, W., Hanson, S., Rahman, M., Salehin, M. (eds) Ecosystem Services for Well-Being in Deltas. Palgrave Macmillan, Cham. https://doi.org/10.1007/978-3-319-71093-8_18
- Yu, J., Li, Y., Han, G., Zhou, D., Fu, Y., Guan, B., . . . Wang, J. (2014). The spatial distribution characteristics of soil salinity in coastal zone of the Yellow River Delta. *Environmental Earth Sciences*, 72(2), 589-599. doi:10.1007/s12665-013-2980-0

Additionally, the previous comments about the relationship between soil moisture and salinity were not centered on measurement alone, and I apologize that this was not more clear. The question about salinity-moisture covariance was about the system in general, as soils closer to the water line should be saltier and also should have more moisture.

Response: We fully agree that the relationship between soil moisture and salinity in coastal areas is complex and dynamic. As mentioned by the Reviewer, soils closer to the seawater line should be saltier and also should have more moisture due to seawater intrusion. When soil moisture drops due to water evaporation, soil salinity will be exacerbating. But the replenishment of fresh water can reduce the salinity of soils. Thus, in this study, we opted to consistently utilize the EC of air-dried soil as the index of salinity. And in the revised manuscript, we have added the relevant content on the ecological significance of soil salinity gradient. Please see the details in the last response.

Specific comments:

L207-208: Please list out the three salt-tolerant mechanisms here (they can be summarized as in Figure 4).

Response: Done. Please see lines 205-206 of the manuscript.

L227: For consistency with Figure 5, C-degradation genes should also be identified as CAZymes here.

Response: Done. Please see line 225 of the manuscript.

REVIEWERS' COMMENTS

Reviewer #1 (Remarks to the Author):

The authors properly addressed the previous comments; however, I have one follow-up on my previous comment:

I see in Table S7 you tested the correlation between several traits with other edaphic properties, but I noticed that genome size is missing. Could you add that to the table? It seems important considering many of the results also hinge on this measurement of genome size.

Otherwise, I only have a few comments:

Fig 3A: Needs units on y-axis label

Fig 3B: There is a small icon over the metabolism boxplot

194: Typo – “Ffatty acid metabolism”

L275: There is no Fig. 8, I’m assuming this should be “Fig. S8”?

Reviewer #2 (Remarks to the Author):

The noteworthiness, significance, validity, methodology, and detail remain unchanged from my previous responses. The flaws I observed in analysis, interpretations and conclusions have been addressed.

REVIEWERS' COMMENTS

Reviewer #1 (Remarks to the Author):

The authors properly addressed the previous comments; however, I have one follow-up on my previous comment:

I see in Table S7 you tested the correlation between several traits with other edaphic properties, but I noticed that genome size is missing. Could you add that to the table? It seems important considering many of the results also hinge on this measurement of genome size.

Response: Many thanks for the suggestion. Please note that Table S7 has been updated to Table S2 due to changes in the order of the tables. Regarding to the genome size data, we have to say it was not missing. Actually, we are not able to obtain the average genome size in each site, but the genome size of each taxon. Thus, direct correlation between genome size and the EC were not presented in Table S2. However, KOs abundance can serve as a good proxy for genome size, given the strong positive relationship between genome size and the number of genes. Our result (Fig. R1) and previous studies (Fig. R2 and Fig. R3) support this notion.

Fig. R1. Relationship between genome size and the overall KOs abundance.

Fig. R2. Relationship between genome size and number of genes in the genome of strain La 6 compared to the genomes of 114 members of the Roseobacter group (from Howat et al., 2018).

Fig. R3. The relationship between genome size and the number of protein-coding genes in the genomes of 21 organisms. The number of protein-coding genes is positively correlated with genome size. (adapted from Howat et al., 2018)

References

Howat AM, et al., Comparative Genomics and Mutational Analysis Reveals a Novel XoxF-Utilizing Methylophile in the Roseobacter Group Isolated From the Marine Environment. 2018. *Frontiers in Microbiology* 9: 766.

Yasui, Y., Hasegawa, E. 2022. The origination events of gametic sexual reproduction and anisogamy. *Journal of Ethology* 40: 273–284.

Otherwise, I only have a few comments:

Fig 3A: Needs units on y-axis label

Response: Units have been added to Fig. 3a, Fig. 4a and Fig. 5b!

Fig 3B: There is a small icon over the metabolism boxplot

Response: The icon has been deleted!

194: Typo – “Ffatty acid metabolism”

Response: Done!

L275: There is no Fig. 8, I’m assuming this should be “Fig. S8”?

Response: it has been replaced with Fig. S8.

Reviewer #2 (Remarks to the Author):

The noteworthiness, significance, validity, methodology, and detail remain unchanged from my previous responses. The flaws I observed in analysis, interpretations and conclusions have been addressed.

Response: we again appreciate the Reviewer’s efforts in evaluating and improving our manuscript.